# A continuum of invariant sensory and behavioral-context perceptual coding in secondary somatosensory cortex

Román Rossi-Pool [1 ✉], Antonio Zainos[1], Manuel Alvarez [1], Gabriel Diaz-deLeon[1] & Ranulfo Romo [1,2,3 ✉]

A crucial role of cortical networks is the conversion of sensory inputs into perception. In the cortical somatosensory network, neurons of the primary somatosensory cortex (S1) show invariant sensory responses, while frontal lobe neuronal activity correlates with the animal's perceptual behavior. Here, we report that in the secondary somatosensory cortex (S2), neurons with invariant sensory responses coexist with neurons whose responses correlate with perceptual behavior. Importantly, the vast majority of the neurons fall along a continuum of combined sensory and categorical dynamics. Furthermore, during a non-demanding control task, the sensory responses remain unaltered while the sensory information exhibits an increase. However, perceptual responses and the associated categorical information decrease, implicating a task context-dependent processing mechanism. Conclusively, S2 neurons exhibit intriguing dynamics that are intermediate between those of S1 and frontal lobe. Our results contribute relevant evidence about the role that S2 plays in the conversion of touch into perception.

[1] Instituto de Fisiología Celular—Neurociencias, Universidad Nacional Autónoma de México, Mexico City, Mexico. [2] Centro de Ciencias de la Complejidad, Universidad Nacional Autónoma de México, Mexico City, Mexico. [3] El Colegio Nacional, Mexico City, Mexico. ✉email: romanr@ifc.unam.mx; ranulfo. romo@gmail.com

Key to understanding the emergence of a percept in the cerebral networks is how sensory inputs are converted into perceptual reports. Is there any cortical area where a sensory representation coexists with a perceptual code? Do these responses appear as a continuum between distinct neural codes? Would these distinct neural codes be associated with separable subnetworks? These questions have been investigated in rodents and primates using different sensory tasks[1–12], showing evidence that some cortical areas may play a relevant role in the conversion of sensory inputs into perceptual responses. However, it has been hard to decode these neural operations across cortices, especially quantifying the degree of sensory or perceptual responses exhibited by a neuron. Vibrotactile discrimination tasks establish an appropriate experimental setting to further explore these questions in behaving monkeys[3,13]. While the temporality of each stimulus is represented faithfully and homogeneously in the primary somatosensory cortex (S1)[14], frontal lobe neurons exhibit complex and heterogeneous responses associated with working memory and perceptual reports[15]. In other words, S1 and the frontal lobe demonstrate disparate signals that correspond to different stages of cognitive processing. The two processing stages may require an intermediary that contains both types of signals, representing sensory inputs for transformation into perceptual reports. Based on proposed hierarchies of the cortical somatosensory network[16–18], that intermediary could be the secondary somatosensory cortex (S2).

Contrary to S1, S2 neurons display large, multi-digit or bimanual receptive fields. Previous anatomical evidence has suggested that S2 is largely connected with downstream, as well as upstream areas[19–25]; this single area could have access to faithful sensory inputs (bottom-up)[4,26], as well as mnemonic information that is solely found in the frontal lobe dynamics (top-down). Further, their neuronal responses could depend on task context[11,27,28]. While a transformation of the sensory code was observed between S1 and S2[14], the coexistence of categorical coding with sensory responses has yet to be studied in S2. The division between neurons representing the sensory inputs and neurons representing the categorical reports has remained unclear in the somatosensory network. Moreover, what is the role of S2 during non-demanding tasks, where S1 responses remain unchanged and frontal lobe coding disappears[3,13,15]? Could S2 act as a switch, transforming sensory information on the basis of task requirements?

In this work, we focused on behavioral conditions in which knowledge of the temporal structure of the stimulus pattern is essential to solve the task. We employed a temporal pattern discrimination task[15] (TPDT) to analyze the neuronal responses recorded in S2. The precise timing of each pulse matters during the TPDT, since the monkeys discriminate between patterns based on their temporal structure. Unlike other somatosensory tasks[15,29,30], an intensive code cannot be used to resolve the TPDT. When we computed the S2 coding dynamics, we found that S2 neurons displayed complex coding associated with the stimuli, early working memory, comparison and decision reports.

We identified activity patterns that mirror the processing stages observed in S1 and in the dorsal premotor cortex (DPC). Focusing on one population extreme, the most sensory S2 neurons showed phase-locked responses to the stimulus and that were invariant to task context and decision outcome; conversely, the responses of S2 perceptual categorical neurons were severely affected during errors, and entirely ablated during a non-demanding task variant (light control task [LCT]). Further, the S2 population reflected a range of intermediate dynamics that varied between pure sensory and pure categorical; the vast majority of the S2 network falls along this continuum of combinations. Moreover, across the S2 responses, categorical

information increased during the TPDT with respect to the LCT, and sensory information diminished. Consequently, which information is predominant in the whole S2 population strongly depends on task context. The entire S2 population demonstrated response and coding latencies that lay between those of S1 and DPC. S2 sensory neurons exhibit significantly longer latencies than area 3b neurons (S1), while S2 categorical neurons display significantly shorter latencies than DPC. Since categorical dynamics emerge first in S2, they are unlikely to originate as a top-down signal from DPC, although we cannot discard other frontal areas as candidate sources. As an extension, we asked if these distinct coding dynamics depended on two separable subnetworks; however, we found neither spatial segregation based on coding dynamics nor timescale differences across S2. Despite the extreme diversity in S2 coding responses, they appear to develop at the same processing stage. Collectively, our findings indicate that S2 is an intermediate processing area where a continuum of neuronal responses, from sensory to categorical, best characterizes the entire population. This suggests that S2 plays a role in the transition from sensory inputs to perceptual behavior.

## Results

**Single-neuron responses during the TPDT.** We trained two monkeys in the TPDT, in which they reported whether two temporal patterns composed of vibrotactile flutter stimuli (P1 and P2) were the same (P2 = P1) or different (P2 ≠ P1)[15] (Fig. 1a, "Methods"). There were two possible temporal patterns: extended (E), which presents 5 pulses periodically, and grouped (G), which presents 3 of the 5 pulses center-grouped. Importantly, stimulus mean frequency (5 Hz) and duration (1 s) were held constant, so the monkey must restrict its discrimination to the period between the initial and final boundary pulses. Thus, the stimuli presented in each trial could be one of 4 possible pairs, or classes: G-G (c1), G-E (c2), E-G (c3), and E-E (c4). The average performance across S2 recording sessions during the TPDT was 84% (±7%), remaining consistent across classes (Fig. 1b).

We recorded extracellular activity from 1646 neurons in S2 (Fig. 1c, "Methods") during the monkeys' performance of the TPDT (Monkey RR17, $n = 1035$; Monkey RR20, $n = 611$). The responses of 12 exemplary S2 neurons are shown in Fig. 1d–f and Supplementary Fig. 1a–i. Contrary to S1[15], S2 neurons displayed a broad repertoire of responses with clearly distinguishable neuronal dynamics. Several neurons were entrained by the stimuli (Fig. 1d and Supplementary Fig. 1a, b), limited to faithful responses tracking the patterns. Another group of neurons exhibited partially phase-locked responses but also encoded some of the task parameters categorically (Supplementary Fig. 1c, d). For example, the neuron of Supplementary Fig. 1f had a much stronger response for G-patterns during both stimulus periods. Supplementary Fig. 1c is phase-locked, however, it diminished its activity during a specific class, c1. These examples reveal that some S2 neurons exhibit intermediate dynamics between pure sensory and pure categorical. On the other hand, some neurons revealed predominantly categorical responses (Fig. 1e, f and Supplementary Fig. 1g–i). These cells do not track the stimulus, suggesting a complete transformation of sensory inputs into abstract categorical representations. This first panorama lets us summarize S2 neurons as consisting of a pure sensory group, a pure perceptual categorical group, and a spectrum of responses mixing both dynamics.

**Single-neuron responses during the TPDT vs. the LCT.** Several of the S2 neurons recorded during the TPDT were also recorded during the LCT ($n = 313$; Monkey RR17, $n = 189$; Monkey RR20, $n = 124$), a control variant of the active task. In each trial, the

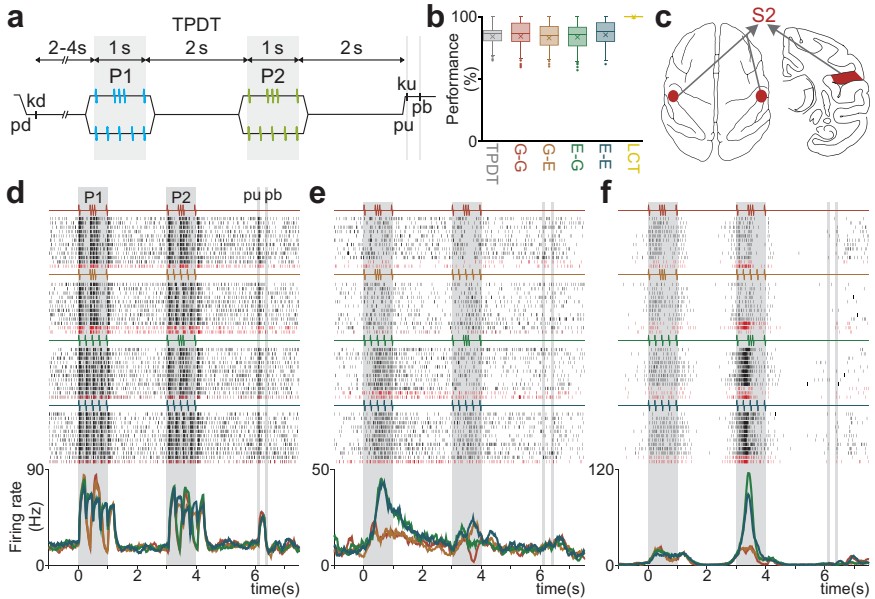

**Fig. 1 Temporal pattern discrimination task (TPDT) and activity of single neurons in S2. a** Trials' sequence of events. The mechanical probe is lowered (pd), indenting the glabrous skin of one fingertip of the right, restrained hand (500 μm); in response, the monkey places its free hand on an immovable key (kd). After a variable prestimulus period (from 2 to 4 s), the probe vibrates for 1 s, generating one of two possible stimulus patterns [P1, either grouped (G) or extended (E); mean frequency of 5 Hz]. Note that in extended pattern (E), pulses are delivered periodically. After a first delay (2 s length, from 1 to 3 s) between P1 and P2, the second stimulus (P2) is delivered, again either of the two possible patterns [P2, either G or E; 1 s duration]; this is also called the comparison period. After a second 2 s delay (from 4 to 6 s) between the end of P2 and the probe up (pu), the monkey releases the key (ku) and presses, with its free hand, either the lateral or the medial push button (pb) to indicate whether P1 and P2 were the same (P1 = P2) or different (P1 ≠ P2). **b** Performance for the whole TPDT (gray, *n* = 423 sessions), for each class [G-G (red), G-E (orange), E-G (green), E-E (blue)] and for the whole LCT (yellow, *n* = 76 sessions). See legend of Supplementary Fig. 5 for box-plot statistics and Supplementary Fig. 5a, b for box-plots and statistics for the individual monkeys. **c** Top of the brain (left figurine) for approaching the secondary somatosensory cortex (S2) and coronal section of the brain (right) for locations of recordings in S2 (red spots). Recordings were made contralateral and ipsilateral to the stimulated fingertip. **d–f** Raster plots of three S2 neurons sorted according to the four possible classes (stimulus pairs). Each row is a single trial, and each tick is an action potential. Trials were interleaved randomly, although the rows were sorted by class afterward (only 10 out of 20 trials per class are shown). Correct and incorrect trials are indicated by black and dark red ticks, respectively. Average firing rates (PSTHs), per class, demonstrated in traces below each raster. Color traces indicate the four possible classes: G-G (red); G-E (orange); E-G (green); and E-E (blue). One neuron exhibits a sensory response (**d**) while the other two exhibit categorical activity (**e–f**).

animals received the same stimuli as in the TPDT, but the correct decision report was guided by a continuous visual cue ("Methods"). As opposed to the TPDT, the performance for the LCT was consistently 100% (Fig. 1b), demonstrating that it was not as cognitively demanding. In a previous work, we observed that neurons in area 3b (S1) do not alter their responses during the LCT[15], although DPC neurons ceased their task-parameter coding[15,31]. Thus, DPC neurons were recruited to code task-relevant information exclusively during the cognitively demanding task (TPDT).

The examples in Fig. 2 and Supplementary Fig. 2 show the responses of ten typical S2 neurons that were tested in both the TPDT and the LCT. Analogous to area 3b, the pure sensory responses are not affected by context (Fig. 2a and Supplementary Fig. 2a, c). On the other hand, intermediate neurons alter only portions of their coding during the LCT (Supplementary Fig. 2b, d, e). The neuron in Supplementary Fig. 2e exhibits G-pattern categorical responses during P2 in the TPDT, but this coding response was lost and only the sensory responses remained during the LCT. In other words, in intermediate neurons, pure sensory responses increased, and categorical responses diminished during the LCT (Supplementary Fig. 2b, d, e, see Supplementary Fig. 8). Further, neurons with clear, or pure, categorical responses (Fig. 2b and Supplementary Fig. 2f) stop coding task parameters. Summarily, S2 neurons modify their categorical responses independently of their sensory responses; the perceptual coding is context-dependent, and the sensory responses are not.

**Context-dependent coding dynamics**. To measure the coding capacities of S2 neurons as a function of time, we employed receiver operating characteristic (ROC) to compare pairs of firing rate distributions associated with each of the four classes (Supplementary Fig. 3). We tested each time bin to identify one of the four coding profiles associated with different task parameters: stimulus pattern identity (P1 or P2), class selectivity, or decision outcome ("Methods"). With this, we were able to calculate the percentage of S2 neurons (*n* = 1646) that coded each task parameter during the TPDT (Fig. 3a). A large percentage of neurons coded the identity of the first pattern (P1, cyan) during the P1 period. The number of S2 neurons coding P1 identity decreases significantly at the beginning of the working memory period, recurring in a smaller proportion at the end of the delay. This reappearance of P1 coding is mainly due to late neurons (see Fig. 1c), potentially serving to recall this information for use during the comparison period. Notably, no S2 neurons coded P1 identity continuously throughout the working memory delay (Supplementary Fig. 4a), unlike the persistent working memory coding demonstrated in DPC (Supplementary Fig. 4b). The comparison period (3 to 4 s) began with the extinction of the P1 signal as a high percentage of neurons coding P2 (green) appeared. Along with P2 coding, a strikingly large number of neurons with class-selective coding (see Supplementary Fig. 1c, g) emerged almost simultaneously (pink). Besides that, a small percentage of S2 neurons exhibited modulation based on decision outcome (black). Surprisingly, the decision signal involves a massive portion of S2 neurons

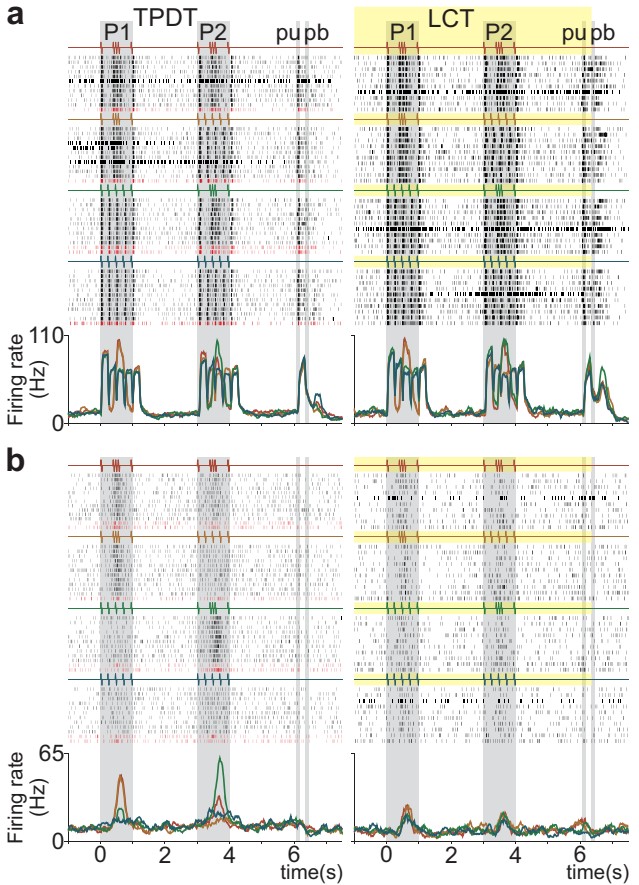

**Fig. 2 Activity of single S2 neurons during the TPDT and LCT. a–b** Raster plots of two additional S2 neurons tested in both tasks: the TPDT (left) and LCT (right). Only the TPDT trial rows are sorted by correct (black ticks) and incorrect (red ticks) trials for each of the four possible classes, individually. There were no errors during the LCT. Traces below the raster plots are class average firing rates per neuron and condition. Each color refers to one class: G-G (red); G-E (orange); E-G (green); and E-E (blue). Sensory responses endured during the TPDT and the LCT (**a**), while categorical responses ceased during the LCT (**b**). Intermediate neurons showed a mix between sensory-invariant and context-dependent categorical responses (see Supplementary Fig. 2).

during the report period (pb, Fig. 3a). Although its role is unclear, this representation was also observed in DPC during the same period[15] (Supplementary Fig. 4e and Fig. 7c).

To what extent are S2 signals dependent on the animal's behavioral report? We applied the same coding scheme (Supplementary Fig. 3) to the S2 population recorded during the LCT ($n = 313$). Notably, the diversity of coding dynamics changed dramatically (Fig. 3b): S2 neurons only coded the stimulus patterns' identity, limited to their respective stimulation periods. Thus, S2 dynamics during LCT are exclusively sensory (Fig. 2 and Supplementary Fig. 2). Further, the total percentage of neurons coding P1 and P2 identity decreased during the LCT; P1 working memory coding, class coding, and decision outcome coding all ceased (Fig. 2 and Supplementary Fig. 2). In addition, note that the decision signal, observed in the TPDT after pu, also disappeared during the LCT (see neurons in Supplementary Fig. 2g, h). Moreover, analogous coding dynamics were observed in the neurons of each monkey during the TPDT and the LCT (Supplementary Fig. 5d–g).

To further quantify these differences, we computed the S2 population instantaneous coding variances ($Var_{COD}$) during both the TPDT (Supplementary Fig. 4c) and LCT (Supplementary Fig. 4d). During the TPDT, coding variance reaches its maximum value during the comparison period, where coding dynamics are most complex. Importantly, S2 $Var_{COD}$ reveals the pure sensory responses in a clearer manner during stimulation. Comparatively, DPC variance dynamics do not exhibit any abrupt peaks related to pure sensory dynamics (Supplementary Fig. 4e vs. c). Moreover, in agreement, $Var_{COD}$ almost vanishes entirely during the middle period of the working memory. In stark contrast, S2 $Var_{COD}$ is only the combination of P1 and P2 stimulus identity variances during the LCT (Supplementary Fig. 4d), each restricted to its respective stimulation period. The decision outcome variance is abolished during the comparison and motor report periods. The maintenance of sensory signals, both in the variance and coding measures computed for S2, is a key characteristic mirroring the dynamics of S1; the ablation of perceptual categorical dynamics in DPC (Supplementary Fig. 4f) and S2 during the control task is a key characteristic of frontal lobe dynamics. Critically, these changes indicate that S2 activity is profoundly related to task context.

**A continuum from phase-lock to categorical neurons.** Afterwards, we employed mutual information[14,32] to isolate neurons with extreme sensory or categorical dynamics. We initially

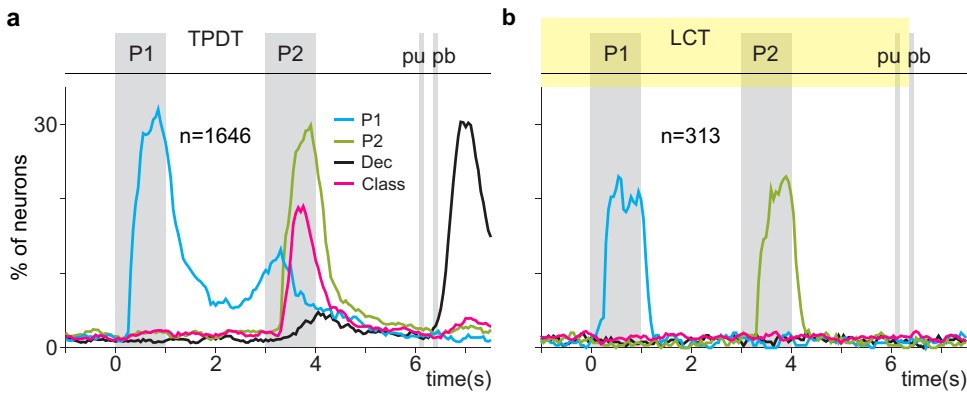

**Fig. 3 S2 Population coding dynamics during the TPDT and LCT. a–b** Percentage of neurons with significant coding (see Supplementary Fig. 3) as a function of time during the TPDT ($n = 1646$) and the LCT ($n = 313$). Traces refer to P1 (cyan), P2 (green), all class coding (pink), and decision coding (black). Note that P1 working memory, decision, and class coding essentially vanished during the LCT: instead, the coding was restricted to stimulus periods in the LCT. Similar to frontal lobe areas, all categorical and perceptual codes are abolished during the control, but akin to S1, S2 sensory responses always persist (see Supplementary Fig. 4).

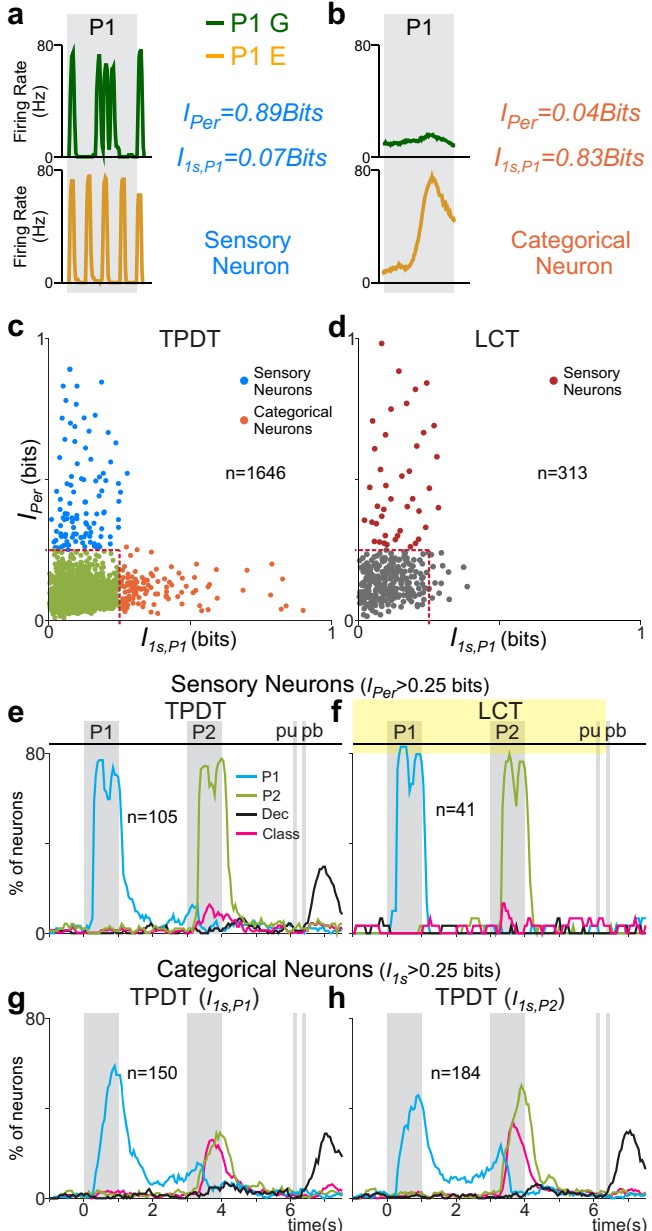

**Fig. 4 Sensory and categorical neurons and their coding dynamics.**
Periodicity mutual information ($I_{Per}$, Eq. (10)) and 1s firing rate mutual information ($I_{1s,P1}$, Eq. (6)) associated with pattern identity were used to identify sensory and categorical S2 neuron subpopulations. **a** Exemplary S2 neuron with high periodicity information during the first stimulation period ($I_{Per}$ = 0.89bits). This neuron demonstrates low values of $I_{1s,P1}$ ($I_{1s,P1}$ = 0.07bits). We can label it as a member of the sensory subgroup. **b** S2 neuron with low periodicity ($I_{Per}$ = 0.04bits) information and a marked categorical response for E pattern. This neuron conveys large values of $I_{1s,P1}$ ($I_{1s,P1}$ = 0.83bits), so it was labeled as categorical. **c** For each S2 neuron recorded during the TPDT ($n$ = 1646), $I_{1s,P1}$ ($x$-axis) is plotted against $I_{Per}$ ($y$-axis), both associated with the identity of P1. Analogous P2 results were omitted. The red dashed lines indicate the arbitrary mutual information criteria ($I$ > 0.25bits) used to label S2 neurons as sensory ($y$-axis) or categorical ($x$-axis). Arbitrary boundaries isolate dynamics features in S2 network based on dominant information value ($I_{Per}$ or $I_{1s,P1}$). Most S2 neurons exhibit low or intermediate values for both $I_{1s,P1}$ and $I_{Per}$ (green points). **d** $I_{1s,P1}$ ($x$-axis) is plotted against $I_{Per}$ ($y$-axis) for each S2 neuron recorded during the LCT ($n$ = 313). Negligibly few neurons exhibit $I_{1s,P1}$ > 0.25bits during the LCT. **e–h** Percentage of each subpopulation of neurons with significant coding as a function of time during the TPDT or the LCT. Traces refer to P1 (cyan), P2 (green), class (pink), and decision coding (black). **e–f** Sensory neurons ($I_{Per}$ > 0.25bits) during the TPDT (right, $n$ = 105) or the LCT (left, $n$ = 41). Most sensory neurons only involved in coding P1 or P2 identity during stimulation periods. **g–h** Categorical neurons computed during the first stimulus ($I_{1s,P1}$ > 0.25bits, $n$ = 150) or second stimulus ($I_{1s,P2}$ > 0.25bits, $n$ = 184) periods. In both cases, P1 coding emerges later and remains longer than in sensory neurons; class coding is present in both categorical neuron types. Decision coding after "pu" is observed in both categorical and sensory neurons during the TPDT. No categorical neurons were identified during the LCT for computing coding dynamics.

identified S2 sensory neurons whose evoked spikes were phase-locked to the stimulus pulses (Fig. 4a). To estimate the degree of periodicity of individual neurons, we computed, for each trial, the frequency power spectrum of their spike trains, during each stimulation period ("Methods"). In neurons with phase-locked responses, the power spectrum should give a high amount of information about stimulus identity[14]. As a result, it should be possible to decode pattern identity based on spike train periodicity in sensory neurons. We calculated the periodicity information during P1 ($I_{Per}$, Eq. (10)), using a permutation test to evaluate significance ($p$ < 0.01, "Methods"). Since analogous results were found using either stimulation period, we chose to show results of the P1 period. Periodicity information is high in the neuron from Fig. 4a ($I_{Per}$ = 0.89bits), but low in categorical neurons without phase-locked responses (Fig. 4b, $I_{Per}$ = 0.04bits). Then, $I_{Per}$ allows us to recognize neurons with strong phase-locking, and putatively sensory, responses. As such, neurons with significant and high $I_{Per}$ (>0.25bits) were classified as sensory. This arbitrary value was set to identify the most extreme S2 sensory neurons. To separate neurons with categorical

responses, we computed the 1s firing rate mutual information associated with the identity of P1 ($I_{1s,P1}$, Eq. (6)) or P2 ($I_{1s,P2}$). Then, $I_{1s,P1}$ (or $I_{1s,P2}$) is blind to any phase-locked response since they produce approximately the same number of spikes for both patterns during the 1s window (Fig. 4a, $I_{1s,P1}$ = 0.07bits). Instead, categorical neurons, that respond differentially for a specific pattern, exhibit high values of $I_{1s,P1}$. As evidence, the differential response to the E-pattern shown in Fig. 4b gives rise to a high value of $I_{1s,P1}$ (0.83bits). Neurons with significant and high $I_{1s,P1}$ values (>0.25bits) were labeled as categorical. Again, although arbitrary, this information criterion allowed us to isolate extreme responses.

Each point in Fig. 4c represents a single-neuron recorded during the TPDT ($n$ = 1646), with its position defined by $I_{1s,P1}$ ($x$-axis) and $I_{Per}$ ($y$-axis). Notice that a comparable number of sensory ($n$ = 105; Monkey RR17, $n$ = 71; Monkey RR20, $n$ = 34) and categorical ($n$ = 150; Monkey RR17, $n$ = 91; Monkey RR20, $n$ = 59) neurons were identified with our criteria ($I_{Per}$ or $I_{1s,P1}$ > 0.25bits, see Supplementary Fig. 6a). Remarkably, no neurons were found along the diagonal that satisfied both criteria (Fig. 4c), which serves as corroboration that pure sensory and categorical neurons represent mutually exclusive dynamics. Neurons not classified as sensory or categorical represented the brunt of the population, exhibiting low or intermediate values for both types of information (green points). Notably, when using the same metrics for the neurons recorded during the LCT ($n$ = 313, Fig. 4d), the plots exhibited drastic changes. In the LCT, almost all neurons with high information were sensory neurons (Fig. 4d); conversely, the population of categorical neurons was drastically reduced. Thus, during the LCT, higher values of $I_{Per}$ were far more common than $I_{1s,P1}$ (Supplementary Figs. 6b and 8a, b). One potential explanation for the two types of dynamics is that

they occur in discretized sub-areas, creating distinct sensory and categorical subnetworks. To address this question, we analyzed the $I_{1s,P1}$ and $I_{Per}$ values conveyed by pairs of neurons recorded together during the TPDT (Supplementary Fig. 7). We found no clusters in the arrangement of S2 neurons; the probability of recording a pair of nearby neurons with pure dynamics was extremely low.

We inquired whether mutual information values depended on the cognitive context (TPDT or LCT), so we compared the same metrics in a subgroup of neurons recorded during both tasks ($n$ = 313, Supplementary Fig. 8). Specifically, we wondered whether single neurons changed the type of information conveyed depending on the task condition. Each neuron represents a point in Supplementary Fig. 8a, defined by the TPDT $I_{Per}$ ($x$-axis) and the LCT $I_{Per}$ ($y$-axis). The angle distribution between the two axes was biased to higher values ($<\theta>$ =57.49°), meaning that neurons have a higher degree of phase-locking responses during LCT than TPDT. In contrast, neurons displayed larger values of categorical information ($I_{1s,P1}$) during the TPDT than the LCT ( $<\theta>$ =31.45°, Supplementary Fig. 8b). Summarily, periodicity information increases during the LCT, while categorical information increases during the TPDT. In agreement, several exemplary neurons with intermediate responses increase their sensory response by decreasing their categorical coding during LCT (Supplementary Fig. 2b–e).

**Sensory vs. categorical coding dynamics**. To elaborate, we analyzed the dynamics at the extremes of neuronal responses, implementing the same coding scheme (Supplementary Fig. 3). In contrast to Fig. 3a, the coding dynamics of sensory neurons increased abruptly and analogously during both stimulus periods (Fig. 4e). Nearly identical coding dynamics were observed during the LCT (Fig. 4f), contrasting sharply with the differences observed between the TPDT and LCT for the whole population (Fig. 3). Distinguishingly, the decision signals after pu were present in sensory neurons during the TPDT, but not during the LCT. Applying variance measures to these sensory neurons produced analogous results (Supplementary Fig. 6c, d). These neurons restrict the majority of their variance and coding dynamics to the stimulation periods.

Conversely, neurons that convey high values of categorical information ($I_{1s,P1}$ or $I_{1s,P2}$) exhibit different coding dynamics (Fig. 4g, h). Several neurons code P1 identity during the early part of the working memory period. In contrast to Fig. 4e, categorical neurons display a high percentage of class coding (pink, Fig. 4g, h). The variance of these neurons (Supplementary Fig. 6e, f) yielded similar features. Neurons with high values of $I_{1s,P2}$ (Supplementary Fig. 6f) depict elevated values of variance during the P2 period, suggesting a preponderant role during the comparison. Importantly, categorical neuron coding increased much more slowly during stimulation than in sensory neurons, and their coding disappeared almost completely during the LCT (Fig. 4d), mirroring the dynamics of DPC (Supplementary Fig. 4e, f).

Returning to the sensory neurons ($I_{Per} > 0.25$bits), the firing patterns of the example sensory neurons (Fig. 1g) were like those observed in the normalized population activity (Fig. 5 and Supplementary Fig. 9). Population averaged responses were entrained to the stimulus patterns, but beyond that, there was no firing rate modulation associated with working memory or decision. When we superimposed the responses (Fig. 5) evoked during hit (Supplementary Fig. 9a) and error TPDT trials (Supplementary Fig. 9b), as well as LCT trials (Supplementary Fig. 9c), we found no statistical differences based on mean squared errors (mse ~1.2–2.6%). From this, we conclude that S2 sensory neurons faithfully tracked the temporal structure of the stimulus patterns, regardless of task context and the monkey's performance.

**Population P1 coding during hit and error trials**. To investigate the degree to which S2 neuron responses correlated with the monkey's choice, we compared the firing rate mutual information associated with P1 during hit vs. error trials. First, we normalized the activity ($z$-score) at each time bin from the 1253 neurons with significant P1 coding (200 ms window, 50 ms step; permutation test, ROC analysis, $p < 0.05$; Supplementary Fig. 3, "Methods"). A 200 ms window was optimal for decoding P1 identity from areas 3b, 1 and S2 sensory neurons (Supplementary Fig. 10). In particular, the information carried by categorical S2 and DPC neurons saturates at this window-width. Later, we split the responses into hit and error trials and measured their P1 mutual information ($I_{P1}(t)$, Eq. (7)). In Fig. 6, we showed $I_{P1}(t)$ during P1 and working memory periods (from 0 to 3 s). Even if most neurons with P1 coding ($n = 1253$), are neither pure sensory ($n = 105$) or categorical ($n = 150$), they are highly informative about P1 identity during hit trials (Fig. 6a, blue), including the early part of working memory. However, during error trials these neurons conveyed less information (Fig. 6a, red).

Each extreme of the S2 responses conveys P1 information with different features. Sensory neurons ($n = 105$; Fig. 6b) conveyed information ($I_{P1}(t)$) analogously during hit (blue) and error trials (red), and only coded sensory inputs during the stimulation period. In the case of categorical neurons ($n = 150$), the results contrast drastically with the sensory group as well as the total population. They demonstrated a slower increase in $I_{P1}(t)$ during hits (Fig. 6c, blue), meaning that P1 information emerged later and then lasted longer, stretching into the beginning of the working memory delay; the most informative point was at the end of the P1 stimulus period. Notably, P1 information carried by categorical neurons almost vanished during errors (Fig. 6c, red). This means that these responses correlated to behavior. The whole population $I_{P1}(t)$ (Fig. 6a) is a combination between sensory and categorical $I_{P1}(t)$, which likely occurs due to the intermediate neurons. In conclusion, activity from sensory neurons does not covary with behavior while categorical responses do; the intermediate responses reflect a dynamical balance.

In Supplementary Fig. 11, we extended these analyses to compute the firing rate mutual information associated with decision and reward. In concordance with Fig. 3, the higher decision signal emerged after the push button press (Supplementary Fig. 11a, b). Notably, S2 neurons carried significant reward information during the period after pb (after 7.5 s, Supplementary Fig. 11c). Hence, it is possible to employ S2 activity to infer if the animal received reward or not. Additionally, this signal emerged subsequently to the categorical decision signal that appeared after pb (Supplementary Fig. 11a, b).

**S2 in the somatosensory hierarchy: latencies and inherent timescales**. Afterward, we calculate response and coding latencies across the S2 population during the TPDT (Supplementary Fig. 12a, "Methods"). We then computed these metrics for the sensory (Supplementary Fig. 12b) and categorical (Supplementary Fig. 12c) neurons. The panels in Supplementary Fig. 12 show the probability distribution for each group. The subgroups can be distinguished by their disparate response and coding latencies; the entire S2 population responds and codes slower than the sensory subgroup (response: 97 ms vs. 33 ms; coding: 432 ms vs. 301 ms), but faster than the categorical subgroup (response: 97 ms vs. 106 ms; coding: 432 ms vs. 477 ms). A clear tendency could be observed in the cumulative curves (Supplementary Fig. 12f);

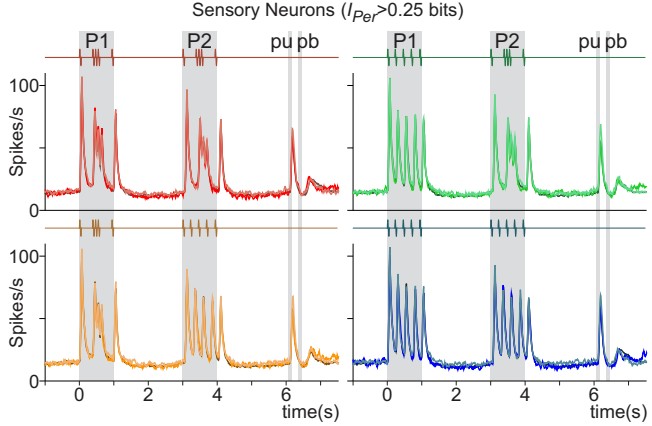

**Fig. 5 Sensory responses during hit, error, and LCT trials.** Sensory S2 neurons selected with significant periodicity information >0.25bits ($I_{Per}$ > 0.25bits, Eq. (10)). Superimposed normalized S2 sensory population activity for hit (dark traces), error ($n = 105$, mid-tone traces), and LCT trials ($n = 41$, light traces). Each color refers to one class: G-G (red); G-E (orange); E-G (green); and E-E (blue). Even if the number of neurons is the same for hit and error responses, the number of error trials is far fewer. Differences between responses associated with each class, calculated using integral square error, were found to be small (from 1.2 to 2.6%). Normalized LCT activity included all trials, since animals have no errors during this control (see extended version in Supplementary Fig. 9).

sensory neurons were the fastest, and categorical the slowest, for S2 neurons. Moreover, the population recorded during the LCT ($n = 313$, Supplementary Fig. 12d) displayed a trend to be faster than the TPDT population for both latency types (response: 72 ms vs. 97 ms; coding: 363 ms vs. 432 ms).

To further explore the S2 population's role in the somatosensory hierarchy, we calculated latencies in other cortex populations (area 3b, S1 and DPC) during the TPDT (Supplementary Fig. 13). As one would expect, 3b neurons responded and coded the fastest of all. However, a remarkably slight difference distinguished the marginally slower sensory S2 latencies from the 3b latencies (response: 33 ms vs. 23 ms; coding: 301 ms vs. 241 ms). Neurons from DPC demonstrate the longest latencies but have comparable coding latencies to S2 categorical neurons (477 ms vs. 484 ms, $p < 0.01$); however, their response latencies tend to be much slower (106 ms vs. 281 ms, $p < 0.001$). Categorical S2 neurons start responding before DPC neurons. Even if the whole S2 neuronal population exhibits intermediate latencies[33], sensory neuron responses resembled those of 3b neurons and categorical coding resembled the slower trends found in DPC.

Moreover, in recent works, timescales of intrinsic fluctuations across cortices were presented within a hierarchical framework[17,34], using the autocorrelation function. We applied this metric to each subgroup and to the entire network of S2 (Supplementary Fig. 14a). Surprisingly, we observed analogous autocorrelation decay rates for the whole S2 population ($\tau = 178$ ms), as well as for sensory ($\tau = 182$ ms) and categorical neurons ($\tau = 187$ ms). Even if sensory and categorical neurons exhibit completely different latencies, their autocorrelation functions are similar. These results support the idea that, although different in function, S2 subpopulations are embedded within the same processing stage (see Supplementary Fig. 7). When extended to S1 and DPC, the same measure established a hierarchical order across cortices (Supplementary Fig. 14b). S1 autocorrelation exhibits a much shorter decay constant ($\tau = 67$ ms), indicating that information reverberates minimally within this network. On the contrary, DPC yields a longer decay constant ($\tau = 182$ ms).

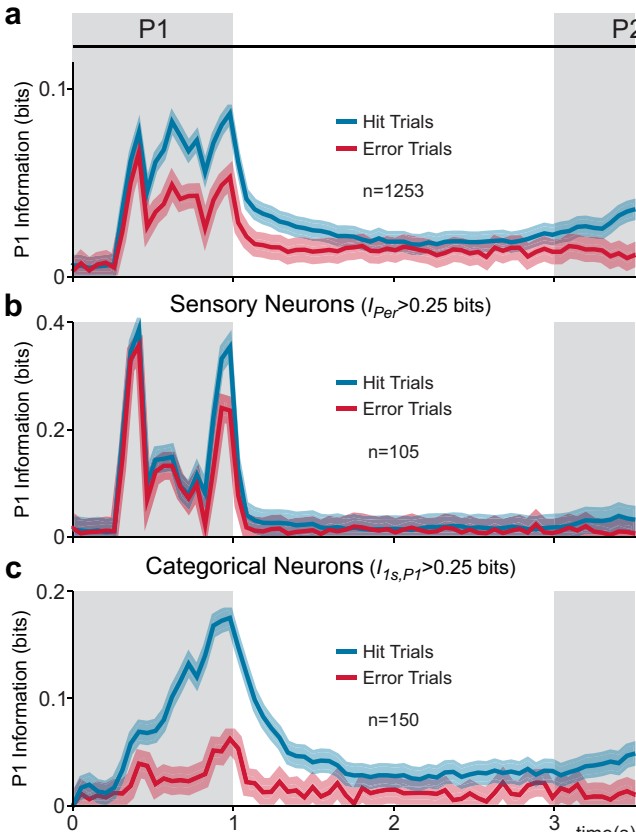

**Fig. 6 P1 mutual information in sensory vs. categorical neurons during hit and error trials.** S2 population firing rate mutual information associated with P1 identity, computed as a function of time ($I_{P1}(t)$, Eq. (7)) during hit (blue) and error (red) TPDT trials. Note that while $I_{1s,P1}$ (Eq. (6)) computed the information at a single time bin that covers the whole stimulus period, $I_{P1}(t)$ (Eq. (7)) measured the information associated with P1 in each time bin (200 ms window with 50 ms steps, "Methods"). **a** Neurons with at least 4 consecutive time bins with significant P1 coding ($n = 1253$) were employed to calculate $I_{P1}(t)$ during hit and error trials. Intriguingly, most of these neurons show intermediate coding, and do not belong to either subgroup. **b** $I_{P1}(t)$ for hit and error trials of the sensory S2 neurons ($I_{Per}$ > 0.25bits, Eq. (10), $n = 105$). As one could observe, the mutual information was nearly invariant during error trials. **c** $I_{P1}(t)$ for hit and error trials of the categorical subgroup of S2 neurons ($I_{1s,P1}$ > 0.25bits, Eq. (6), $n = 150$). The P1 mutual information in hits increased slower than for sensory neurons and decreased drastically in errors. Shadows indicate the information confidence intervals at 95% estimated through bootstrap technique.

Notably, all autocorrelation functions were unaffected during the LCT (Supplementary Fig. 14c). Even though coding dynamics may change completely during the LCT (Fig. 7), their autocorrelation functions do not.

**Dynamical coding across somatosensory areas during the TPDT and LCT.** To better contextualize our results, we computed the coding dynamics (Supplementary Fig. 3) for 3b and DPC exactly as we did for S2 (Fig. 7). Coding dynamics changed completely across cortices during the TPDT (Fig. 7a–c). While area 3b activity is only involved in P1 or P2 coding during stimulation periods (Fig. 7a), S2 and DPC display much more complex dynamics. Note that in DPC, P1 coding is persistent throughout the working memory delay (Fig. 7c and Supplementary Fig. 4), and class and decision coding persist through the

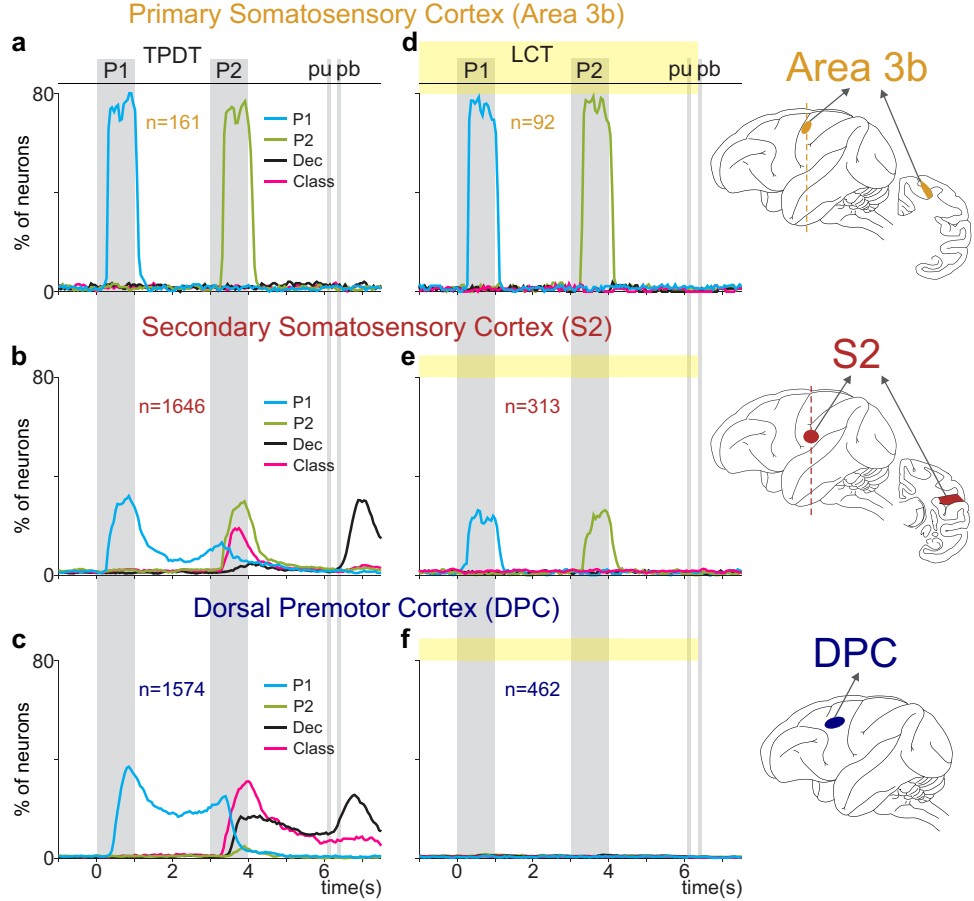

**Fig. 7 Population coding dynamics across the cortex during the TPDT vs. the LCT.** Percentage of neurons with significant coding (Supplementary Fig. 3) as a function of time during the TPDT (right, **a**–**c**) or the LCT (left, **d**–**f**). Traces refer to P1 (cyan), P2 (green), class (pink), and decision coding (black). Neurons were recorded from different cortical areas: Area 3b (TPDT [**a**, $n = 161$] and LCT [**d**, $n = 92$]), S2 (TPDT [**b**, $n = 1646$] and LCT [**e**, 313]), and DPC (TPDT [**c**, $n = 1574$] and LCT [**f**, $n = 462$]). Note that area 3b is included in the primary somatosensory cortex (S1). The same y-axis scale (0 to 80%) was used across panels to facilitate comparison. Area 3b demonstrates an entire network commitment to representing stimulus. In S2, the sensory representation diminishes and the categorical representation emerges. The emergent categorical representation is, in turn, the only activity code observed for DPC.

second delay. In contrast to S2 (Fig. 7b), there is no P2 coding in DPC, meaning no pure sensory responses were identified. During the comparison period, DPC neurons focus almost completely on class and decision coding[33]. Further, analogous to 3b (Fig. 7d), sensory S2 neurons remained invariant during the LCT (Fig. 4f). However, DPC and S2 coding changed completely (Fig. 7e, f). In DPC, all coding disappeared entirely during the LCT. This is further evidence that invariant sensory responses are not present in DPC. While S2 depicts both sensory-invariant and categorical perceptual responses, area 3b and DPC only exhibit one. Therefore, S2 might act as a switch network that allows information flow based on necessity, while potentially aiding in the sensory input transformation itself.

## Discussion

Our work sought to characterize the S2 neural responses during a temporal pattern discrimination task. We show conclusive evidence that a duality exists in S2 between sensory and categorical coding. On one hand, a percentage of S2 neurons constitute a sensory subgroup that is invariant to both behavior and cognitive demand of the TPDT. Alternatively, another specialized group of neurons categorically encode the stimulus identity with a clear dependence on task context (TPDT or LCT) and behavioral reports (hits or errors). Importantly, the information conveyed by

the S2 population exhibits a context-dependent shift: during LCT, categorical coding essentially vanished while sensory responses prevailed. Moreover, we employed coding dynamics, latencies and autocorrelation timescales to frame the intermediate behavior of S2 within the somatosensory hierarchy.

During vibrotactile frequency discrimination and detection tasks, S2 neurons can exhibit categorical responses highly correlated with the monkeys' decision[16,18,33]. In contrast with these two tasks, the precise internal structure of the stimulus is necessary for the temporal pattern discrimination. Then, pure sensory and categorical S2 neurons may both be required during the TPDT. However, the vast majority of neurons exhibit a mix of dynamics. We explored the role of these intermediate neurons with a cognitively non-demanding control task, the LCT. Notably, the entire spectrum of neurons shifts the information conveyed depending on the task's cognitive demand. During the LCT, the sensory information is more abundant, but categorical information predominates during the TPDT. We speculate that the dual role of S2 neurons may be crucial in transforming pure sensory signals into more abstract, categorical codes.

Importantly, several neurons in S2 sustained their response coding during the early delay period between P1 and P2. This early working memory coding is markedly affected or entirely lost in error trials. Thus, while the sensory inputs are the same, the categorical transformation is directly correlated with the

monkeys' behavioral reports. In contrast to DPC, no S2 neurons exhibit persistent coding during the delay[14,33]. Congruently, a disappearance of persistent coding and a recall signal was recently identified in mice S2 neurons during working memory[11]. Further, S2 inactivation during this recall deteriorated performance. We speculate that this information is recalled from higher-order cognitive areas, such as DPC. This frontal lobe top-down signal may play a fundamental role during the comparison of sensory information in S2.

The comparison of latencies between cortical areas and neuron subgroups yielded a great deal of information. The entire S2 population responds and codes with different temporal lags depending on each neuron's coding profile, but all fall between those of 3b and DPC. The fastest portion of the S2 network lies at the sensory extreme and the slowest at the categorical extreme, paralleling the disparate 3b and DPC populations. Note that S2 appears to be located conveniently to receive both bottom-up and top-down inputs[20–25]. Importantly, several works have suggested that S1 acts as a driver in the processing role of S2[19,20,24], while others suggest that the somatosensory thalamus (VPL) plays the driver role instead[23]. Our results show strong evidence that a subgroup of the S2 population responds similarly to 3b neurons, with slightly longer average response latencies (33 ms vs. 23 ms). Moreover, the slowest neuron still responds and codes faster than DPC, so all dynamics appear initiated in S2 before DPC is recruited. With regards to other frontal areas (i.e., MPC, VPC, PFC), prior studies in other tasks have shown that their latencies are analogous in all cases to those found for DPC[18,33].

These results suggest that S2 may not be receiving the initial categorical information from the frontal lobe. This is supported by the fact that DPC does not contain considerable P2 information, which would be necessary to compute categorical (class or decision) representations through stimuli comparison. S2, on the other hand, does contain information about P2, but it lacks the persistent representation of P1. We are thus presented with a possible functional loop between these parietal cortices and the frontal lobe. The frontal lobe stores information that will be useful for categorical abstraction; but this is, in turn, relayed back to S2 so as to converge with the sensory input of P2, permitting the actual computation. Afterwards, the result is sent back.

Motivated by the clear differences between sensory and categorical responses, we wondered whether these contrasting dynamics arise within differentiable subnetworks. It is unlikely to record pairs of pure sensory or categorical neurons in nearby electrodes, indicating that neurons with similar responses do not appear in spatial clusters. Further, the intrinsic timescale of neural fluctuations, estimated with the autocorrelation decay constant, increases from sensory to frontal lobe cortices in monkeys[17,34] and mice[35]. We found the same timescale for all S2 responses, regardless of dynamic profile. All this evidence taken together does not support the notion of subnetworks or structural differences. Besides, across the somatosensory hierarchy, S1 displays fast timescales and phase-locked responses, while S2 and DPC exhibit much longer time constants that are appropriate for temporal integration. Further, the optimal integration window to decode pattern identity from 3b and S2 sensory responses was ~200 ms, which concords with the S2 timescale found here. Moreover, categorical S2 and DPC neurons saturate their coding capacity at around the same window-width.

Recently, we have analyzed the heterogeneous responses observed in frontal lobe neurons with dimensionality reduction techniques[3,31,36], which allow us to condense the network's signals, preserving the significant population dynamics[37–39]. Notably, for each component, the weights given to each neuron occurred with a Gaussian-like distribution[31,36]. These continua of responses are parallel to the gradient of intermediate dynamics we

have shown in S2, so a promising line of inquiry would be the application of such techniques to interpret mixed responses at a population level. In another recent study, analogous methods were applied to investigate the simultaneous recordings of two visual sensory areas, V1 and V2[40]. They revealed a population level mechanism in V1 that influences a small part of the total activity fluctuations observed in V2. The somatosensory network could be implementing a similar mechanism for the routing of sensory information from S1 to S2 and categorical information to DPC. Future experiments and analyses are required to address this hypothesis.

We would like to highlight the high percentage of S2 neurons that code the decision during and after the pb event. The loss of this signal during the LCT means that it could not be associated with motor execution. Although the functional purpose of this decision coding is not evident, we hypothesize that it could be associated with the S2 reward signal observed afterwards. One possibility is that both signals may be necessary for network rewiring, employing the choice outcome to adapt future decisions[41,42]. Further, in a recent model, activity surpassing a threshold leads to an ignition, causing a distribution of information across cortices[43,44]. If subjects do not attend to the stimulus, the ignition may fail. In terms of our S2 findings, we speculate that this area could play a relevant role in the ignition-gated distribution of categorical information to frontal areas[3]. Furthermore, unpublished results from our lab have suggested that animals with lesions applied to S2 are no longer able to perform the task adequately, similar to the effect observed from lesioning S1[45]. Ultimately, this could mean that S2 is necessary for cognitive processing within the cortical network[46,47].

To conclude, both sensory and categorical responses were found within S2. While categorical responses covaried with behavior and ceased during the non-demanding task (LCT), the sensory responses prevailed. The information conveyed by the network depends on context, with categorical information dominating during the active task and sensory information during the control task. From this, we speculate that this area may play a fundamental role in the conversion of sensory inputs to more abstract, conceptual and categorical responses. Therefore, S2 may act as a switch network: always receiving the same sensory inputs, but selectively converting and transmitting abstract representations when the task demands it. This may be a central processing principle, not only for S2, but also for other areas related to other sensory tasks and modalities.

## Methods

**Temporal pattern discrimination task (TPDT).** The TPDT used here has been previously described[15]. In brief, two monkeys (*Macaca mulatta*) were trained to report whether the temporal structure of two vibrotactile stimuli patterns (P1 and P2) of equal mean frequency (5 Hz, 5 pulses) were the same (P2 = P1) or different (P2 ≠ P1; Fig. 1a). The temporal structure of each pattern was either grouped (G) or extended (E) with a fixed stimulation period of 1 s. The five pulses were delivered periodically during the extended pattern (E), and three grouped centered pulses with a smaller distance between them as compared to the first and final pulses, during the grouped pattern (G). Monkeys performed the task in blocks of trials in which the two stimulus patterns had a fixed mean frequency. The right arm, hand and fingers were held comfortably but firmly throughout the experiments. The left hand operated an immovable key (elbow at ~90°) and two push buttons in front of the animal, 25 cm away from the shoulder, at eye level. Stimuli were delivered to the skin of one digit from the distal segment of the right, restrained hand via a computer-controlled stimulator (2 mm round tip, BME Systems, Baltimore, MD). The initial event marks the beginning of the trial by descending the probe to a skin indentation of 500 μm (probe down, "pd" in Fig. 1a). Vibrotactile stimuli consisted of trains of short mechanical pulses; each pulse consisted of a single-cycle sinusoid lasting 20 ms. Time is always referenced to first stimulus onset (0 s corresponds to the start of P1). In a trial, P1 and P2 were delivered consecutively to the glabrous skin of one fingertip, separated by a fixed inter-stimulus delay period of 2 s (1 to 3 s). Each stimulus could be one of the two possible patterns: grouped (G, upper trace of Fig. 1a) or extended (E, lower trace of Fig. 1a) pulses. Therefore, in total there were four possible P1-P2 combinations, denominated as classes: G-G (class 1,

c1), G-E (class 2, c2), E-G (class 3, c3) and E-E (class 4, c4). These were presented in pseudo-random order to the monkeys across trials. The monkeys were asked to report whether P2 = P1 (match: combinations E-E and G-G) or P2 ≠ P1 (non-match: combinations E-G and G-E) after a fixed delay period of 2 s (4 to 6 s) between the end of P2 and the mechanical probe rising from the skin (probe up event, "pu" in Fig. 1a). The "pu" was the go signal that triggered the animal's release of the key ("ku" in Fig. 1a). The monkey indicated their decision by pressing one of two push buttons with the left hand ("pb" in Fig. 1a, lateral push button for P2 = P1, medial push button for P2 ≠ P1). As the two stimulus patterns had equal mean frequency over their full duration (1 s), the decision had to be based on comparison of their temporal structure. The animal was rewarded for correct decisions with a drop of liquid. Animals were handled in accordance with standards of the National Institutes of Health and Society for Neuroscience. All protocols were approved by the Institutional Animal Care and Use Committee of the Instituto de Fisiología Celular, Universidad Nacional Autónoma de México.

**Light control task (LCT).** During this control task, events proceeded exactly as described above and in Fig. 1a, except that when the probe touched the skin ("pd"), one of the two push buttons was illuminated, indicating the correct choice. Identical stimuli were used. The monkey grasped the key until the probe was lifted, but in this case the light was turned off when the probe lifted from the skin. The monkey was rewarded for pressing the illuminated button. Maintaining stimuli and arm movements identical to the TPDT, the decision must be based on the visual stimuli instead.

**Task design and performance.** The TPDT is not a simple variation of the vibrotactile frequency discrimination task (VFDT)[33]. Some cognitive demands and the basic structure of the tasks are similar: both require attention to two separate vibrotactile stimuli (TPDT: P1, P2; VFDT: f1, f2), working memory and a comparison to reach the decision report. Nevertheless, the TPDT requires a very different evaluation of the stimuli; as they only differ by their temporal structure, any computation must be restricted to the internal structure to identify, categorize and distinguish between them[15]. Further, the comparison process is significantly different between the two tasks. Expanding on the necessitated computation, the VFDT can be solved by computing a difference between the parametric representation of the stimulus frequencies to indicate whether f1 > f2 or f1 < f2, whereas the TPDT offers no comparable method of solution (in any trial P1 and P2 always have the same mean frequency). The TPDT demands a match (P2 = P1) vs. non-match decision (P2 ≠ P1). Hence, the comparison employs categorical representations (instead of parametric) of the stimulus patterns.

We computed the average performance across S2 recording sessions ($p = 84.0\%$; Monkey RR17, $p = 84.5\%$ and Monkey RR20, $p = 83.1\%$). Fig. 1b and Supplementary Fig. 5a–c). Although each animal received around two years of training, this task was difficult enough to impede 100% performance; this reflects the very high-cognitive demands of the TPDT. To provide some context, the average training period to achieve similar performance levels for the VFDT was about six to eight months;[33] for the vibrotactile detection task[30], the average time was two months. After training in the TPDT, the monkeys saturated their average performance around 84% (Fig. 1b and Supplementary Fig. 5a–c, $n_{SES} = 423$ recording sessions; Monkey RR17, $n_{SES} = 281$; Monkey RR20, $n_{SES} = 142$). In addition, the performance was statistically identical for each class[15]. Notably, task repetition across recording sessions did not improve performance. However, the performance for the LCT was consistently 100% (Fig. 1b and Supplementary Fig. 5a–c, $n_{SES} = 76$ recording sessions; Monkey RR17, $n_{SES} = 49$; Monkey RR20, $n_{SES} = 27$); this reflects the lack of cognitive demand required for the guided-task, as intended by design. As a final observation, the animals were first trained in the LCT, and then gradually introduced to the TPDT. During the recording sessions in S2 (Fig. 1c), animals switched between performing the TPDT and the LCT.

**Recordings.** Neuronal recordings were obtained with an array of seven independent, movable microelectrodes (2–3 MΩ)[16] inserted into S2 (Fig. 1c), either contralateral (left hemisphere) or ipsilateral (right hemisphere) to the stimulated hand. We were careful to record just above the primary auditory cortex (A1), and we tested this using auditory stimuli to ensure that the neurons were only responding to vibrotactile stimuli. The receptive fields of the recorded neurons were all very large and some were bimanual, and since the monkey's hand was carefully fixed in the same manner during each recording session, we do not believe it is possible for these neurons to be responding to motor data in a categorical manner, as would be seen in the parietal ventral area (PV). Concurrently, categorical decision responses during P2 or after pb disappeared during the LCT (Fig. 4).

We collected neuronal data in blocks using different mean frequencies[15]. However, for the analysis described below we will focus on the neuronal responses with the stimulus set illustrated in Fig. 1a (5 Hz). In general, we recorded 20 trials per stimulus pair (c1; c2; c3; c4). Recording sites changed from session to session; the locations of the penetrations were used to construct surface maps in S1, S2 and DPC by marking the edges of the small chamber (7 mm in diameter) placed above each area. It is important to emphasize that the sensory and categorical neuron subgroups were both recorded across the entire S2 region. The probability of recording two pure subgroup responses together is extremely low (Supplementary

Fig. 7), so we did not record sufficient pure pairs for further analyses. In area 3b (S1), we recorded neurons with cutaneous receptive fields confined to the distal segments of the glabrous skin of one fingertip of digits two, three or four, such that the receptive field always corresponded to the stimulated digit. All recordings in DPC were made in the hand/arm region F2. This region is in front of M1 (F1), lateral to the central dimple, posterior to F7 and the genu of the arcuate sulcus[15,33]. The neuronal recording protocol was identical for both the TPDT and LCT.

**Datasets.** We recorded 1646 S2 neurons using the TPDT stimulus set with 5 Hz mean frequency (Monkey RR17, $n = 1035$; Monkey RR20, $n = 611$). Additionally, we have a dataset of $n = 313$ neurons (Monkey RR17, $n = 189$; Monkey RR20, $n = 124$) that were tested in both the LCT and TPDT using the 5 Hz mean frequency set. These neurons were used to compare periodicity and categorical firing rate information between the cognitively demanding TPDT and the guided LCT (Supplementary Fig. 8).

For each neuron of the datasets ($n = 1646$ and $n = 313$), we calculated a time-dependent firing rate per trial using a 200 ms deterministic square kernel with 50 ms steps, beginning 1 s before stimulus pattern P1 and continuing until the end of the trial (1.5 s after the push button press). In Supplementary Fig. 10, we show that this window-width is optimal for decoding pattern information. Importantly, each dataset is defined by four dimensions: N, number of neurons; C, stimulus conditions (classes, always 4); T, time ($-1$ to 7.5 s, always 170 bins); K, number of hit trials (for each class). Further, we constructed a similar dataset with error trials for the 5 Hz TPDT stimulus set. Each recorded neuron had on average 2.9 error trials for a given class. A remarkable feature of this task design is the low number of stimulus conditions (four classes), which were equally demanding for the subject. This design allowed us to have, on average, 15.3 hit trials (and 2.9 error trials) per stimulus class for each studied neuron.

**Single-neuron coding.** This analysis was designed to quantify whether the activity of single S2 neurons was modulated as a function of time by the four stimulus classes used in the task: c1 (G-G); c2 (E-G); c3 (E-G) and c4 (E-E). We employed the same coding scheme used previously to identify single-neuron coding in DPC and S1[15].

Employing only hit trials, we constructed a neuron firing rate distribution for each class. At each time bin we used the ROC to identify class-differential responses; using these class firing rate distributions, we computed the area under the ROC curve (AUROC value) for the six possible class comparisons: c1 vs. c2; c1 vs. c3; c1 vs. c4; c2 vs. c3; c2 vs. c4; and c3 vs. c4. To determine significant AUROC values, we performed a permutation test by randomly shuffling the class labels across trials, while re-computing the AUROC values with the shuffled trials. If the unshuffled AUROC value (≠0.5) reached or exceeded the 95% of the distribution obtained from 1000 shuffled surrogates, responses for the two compared classes were labeled statistically different ($p < 0.05$); otherwise, they were labeled as equal. We should emphasize that statistical equality means that there is not enough neuronal response information to differentiate the two distributions; this does not mean that both distributions were the same.

From this, we produced a library of binary words; for each 200 ms bin we had six digits resulting from the six comparisons (Supplementary Fig. 3). In this coding scheme, the 0's are as important as the 1's. The criterion to assign both was very strict: to avoid random assignments at each time window, we only assigned a binary label of statistical equality (0) or inequality (1) if the same digit was kept for at least four consecutive bins, otherwise no label would be assigned, and that time bin was excluded from the classification. This part of the coding scheme was designed to correct for multiple comparisons. It is important to note that for each time bin this procedure generates a unique code for each neural response, one of our "binary words". However, we isolated four relevant response or coding labels (P1, P2, class and decision) from the 64 binary words. These four profiles are explained below. From 64 binary words, we isolated 7 associated with our labels, while the rest represent mixed or ambiguous codes (Supplementary Fig. 3). Using the binary words computed from the six AUROC values as described above, each time bin was tested for classification into one of four possible coding profiles during the TPDT and LCT.

*P1 coding.* This profile applied to responses that tracked the identity of the P1 pattern. In this case, the responses must be similar for classes c1 (G-G) and c2 (G-E), and for c3 (E-G) and c4 (E-E), which have the same P1, but must differ between all other class comparisons, which have different P1 patterns (Supplementary Fig. 3).

*P2 coding.* As described above, but for responses that tracked the identity of P2. Responses must be similar for c1 and c3, and for c2 and c4, which have identical P2, and must be different for all other class combinations, which have different P2 patterns.

*Class-selective coding.* This profile corresponds to neurons that responded preferentially to one of the four classes. Time bins were labeled according to the class that selectively evoked a response. We associated four binary words with this profile, pursuant to a single rule: the preferred class evoked a unique response,

while the three non-preferred classes were indistinguishable between each other (Supplementary Fig. 3).

*Decision coding.* In this profile, responses must be similar for classes c1 (G-G) and c4 (E-E), as well as c2 (G-E) and c3 (E-G), which share the same outcomes (either P1 = P2 or P1 ≠ P2) and differ for all other class comparisons with distinct outcomes (Supplementary Fig. 3).

Time bins where the six comparisons did not fit any of the binary words described above were considered to be non-coding. Further, to consider that a neuron had significant coding, a minimum of 4 consecutive bins must maintain the same profile. Applying this procedure across all neurons allowed classification of encoding dynamics as functions of time (Figs. 3a, b, 4e–h, 7a–f, Supplementary Figs. 4a, b, 5d–g). This coding scheme rendered two advantages: (1) being able to quantitatively assess and describe all the possible neural codes during all task epochs, and (2) generating coding types that would not overlap in their meaning.

**Instantaneous coding variances across the population.** For each neuron, we averaged the time-dependent firing rate of hit trials per class (c1, c2, c3 or c4). Using the peri-stimulus time histogram (PSTH) of each neuron, we constructed pseudo-simultaneous population responses by combining neural data mostly recorded separately. For each time and class, the population response is defined by an N-dimensional vector in which each component represents the firing rate from a different neuron. This means that including all the recorded neurons ($n = 1646$), we obtained a 1646-dimensional firing rate vector that depended on the time and class ($\bar{r}(t, c)$). The population firing rate averaged over all hit trials ($\bar{r}(t)$) was an N-dimensional vector that measures the mean response for each neuron ($r^i(t)$)as a function of time. For the LCT control condition, the population response was a 313-dimensional firing rate vector.

At each time point, the population instantaneous coding variance ($\text{Var}_{\text{COD}}$, Supplementary Figs. 4c–f and 6c–f, blue trace) was computed as the quadratic square sum of the firing rate fluctuations among classes and neurons:

$$\text{Var}_{\text{COD}}(t) = \frac{1}{N}\frac{1}{4}\sum_{i=i}^{N}\sum_{c=1}^{4}\left(r^i(t,c) - r^i(t)\right)^2 \quad (1)$$

This metric, normalized per neuron, measures the population's variation of firing rate between classes at each time point. In this case, $\text{Var}_{\text{COD}}$ will be associated with any class-related change in the population activity and to stochastic fluctuations (residual noise).

To evaluate the influence of each kind of coding on $\text{Var}_{\text{COD}}$, we calculate the instantaneous variance associated with each task parameter. At each time bin, the population instantaneous P1 variance ($\text{Var}_{\text{P1}}$, Supplementary Figs. 4c–f and 6c–f, cyan trace) was computed as the quadratic square sum of the firing rate fluctuations among P1 identity and neurons:

$$\text{Var}_{\text{P1}}(t) = \frac{1}{N}\frac{1}{2}\sum_{i=i}^{N}\sum_{\text{P1}=1}^{2}\left(r^i(t,\text{P1}) - r^i(t)\right)^2 \quad (2)$$

Analogously, the population instantaneous P2 variance ($\text{Var}_{\text{P2}}$, Supplementary Figs. 4c–f and 6c–f, light green trace) measures the firing rate fluctuations among P2 identity and neurons:

$$\text{Var}_{\text{P2}}(t) = \frac{1}{N}\frac{1}{2}\sum_{i=i}^{N}\sum_{\text{P2}=1}^{2}\left(r^i(t,\text{P2}) - r^i(t)\right)^2 \quad (3)$$

The population instantaneous decision variance ($\text{Var}_{\text{DEC}}$, Supplementary Figs. 4c–f and 6c–f, black trace) measures the firing rate fluctuations of decision identity and neurons:

$$\text{Var}_{\text{DEC}}(t) = \frac{1}{N}\frac{1}{2}\sum_{i=i}^{N}\sum_{\text{DEC}=1}^{2}\left(r^i(t,\text{DEC}) - r^i(t)\right)^2 \quad (4)$$

The value of $\text{Var}_{\text{COD}}$ during the period immediately before P1 onset represented the inherent stochastic fluctuation (residual noise) in the firing rate estimates (~2[sp/s]2); to be interpreted as a degree of population coding, $\text{Var}_{\text{COD}}$ should be higher than this resting-state variance (basal variance). The same reasoning applies to the other specific variances. Accordingly, $\text{Var}_{\text{COD}}$ and $\text{Var}_{\text{P1}}$ depart from their basal values at the same time bins (Supplementary Figs. 4c–f and 6c–f). Further, the times at which any of the specific variances depart from their basal value coincide with the emergence of significant coding in individual neurons (compare Fig. 3a with Supplementary Fig. 4c and Fig. 7c with Supplementary Fig. 4e).

**Sensory population response.** To describe the sensory population responses of S2 ($n = 105$), we normalized the firing rates for each time bin (50 ms window displaced every 10 ms) using the z-score transform. The z-score was computed by subtracting from each trial (hit, error, and control trials) the mean firing rate and dividing the result by the standard deviation (SD) at each time window. The mean and SD for each neuron were calculated using the recorded firing rate activity in hit trials from all time bins in the interval from −1 to 7.5 s of the task. We calculated a mean z-score value for hit, error and control (LCT, $n = 41$) trials for each class to obtain an average sensory population response as a function of time. Finally, we transformed back the mean population z-scores to show responses in terms of firing rates instead of z-scores (Fig. 5 and Supplementary Fig. 8). Back

transformation was computed using the average firing rate values and SDs from all sensory neurons.

**Firing rate information.** Using the firing rate values, we measured their association with P1 and P2 in terms of Shannon's mutual information:

$$I = \sum_{r,\text{P}} P(\text{P})P(r|\text{P})\log_2\left(\frac{P(r|\text{P})}{P(r)}\right) \quad (5)$$

Here, the information ($I$), measured in bits, quantifies the accuracy with which the neural response (the firing rate $r$) can be used to determine the identity of the stimulus pattern (P). The expression $P(r)$ corresponds to the probability of observing a response ($r$) regardless of the stimulus pattern; it was computed using the firing rate probability distribution from all hits during the same time window. $P(\text{P})$ represents the probability that the stimulus pattern takes a value P (G or E), considering only hit trials. $P(r\,|\,\text{P})$ is the conditional probability of observing a response $r$ given a specific stimulus pattern P.

Importantly, to calculate the categorical information, we employed 1000 ms windows that covered the whole first stimulus (from 0 to 1 s) or the whole second stimulus (from 3 to 4 s) period. Then, we quantified the decodable firing rate information conveyed by each neuron about pattern identity (G or E) during P1 or P2, employing a 1 s integration window ($I_{1s}$):

$$I_{1s} = \sum_{r_{1s},\text{P}} P(\text{P})P(r_{1s}|\text{P})\log_2\left(\frac{P(r_{1s}|\text{P})}{P(r_{1s})}\right) \quad (6)$$

Note that $I_{1s}$ in neurons that are tightly phase-locked to the stimulus pulses (phase-locked or sensory neurons), should be near zero (Fig. 4a). Since the number of pulses is the same for each type of pattern G and E, if each pulse is represented equally by a sensory neuron, the firing rate during the whole stimulus period (1 s) should be approximately the same. This means that 1 s-firing rate mutual information ($I_{1s}$) associated with the pattern identity is near zero for sensory neurons. Contrary to that, categorical neurons should have higher values of $I_{1s}$ (Fig. 4b), where pulses generated different responses depending on the pattern identity (G or E).

In Fig. 6, we computed the firing rate mutual information associated with the identity of P1 during hit or error trials for different subpopulations of S2 neurons. We z-scored the 200 ms (see Supplementary Fig. 10) firing rate responses from each hit or error trial at each time bin. Then, we joined the z-score values from different neurons to calculate the population z-score conditional probabilities $P((z(t)|\text{P1}))$ associated with each pattern (E or G). Note that we constructed different distributions for hits and errors. Next, we used the z-score population probabilities to estimate, per time bin, the mutual information associated with P1 during hits or errors:

$$I_{\text{P1}}(t) = \sum_{z(t),\text{P1}} P(\text{P1})P(z(t)|\text{P1})\log_2\left(\frac{P(z(t)|\text{P1})}{P(z(t))}\right) \quad (7)$$

Analogously, we calculated the population firing rate mutual information associated with the decision identity during hit or error trials (Supplementary Fig. 11b). As before, we computed the z-score normalization to the 200 ms firing rate responses, splitting hit and error trials. Then, we constructed population probability distributions associated with decision identity (P1 = P2 or P1 ≠ P2) during hit or error trials:

$$I_{\text{Dec}}(t) = \sum_{z(t),\text{Dec}} P(\text{Dec})P(z(t)|\text{Dec})\log_2\left(\frac{P(z(t)|\text{Dec})}{P(z(t))}\right) \quad (8)$$

Finally, we estimated the firing rate mutual information associated with reward (Supplementary Fig. 11c). In this case, we computed a distribution with all hit trials and another with all error trials. We employed these two population probability distributions to calculate the amount of information associated with the reward, conveyed in the firing rate of the population:

$$I_{\text{Rew}}(t) = \sum_{z(t),\text{Rew}} P(\text{Rew})P(z(t)|\text{Rew})\log_2\left(\frac{P(z(t)|\text{Rew})}{P(z(t))}\right) \quad (9)$$

In Supplementary Fig. 10, we employed different sliding window-widths (from 10 to 1000 ms), moving in 10-ms steps, and quantified the information conveyed by each neuron about pattern identity (G or E) during P1 or P2 for each window-width. Averaging across time points, for each neuron, we computed the mean information values for P1 and P2 as functions of window-width. Finally, we averaged the pattern information values from all neurons to obtain the mean population information for each window. We showed that a 200 ms window is optimal for decoding pattern identity from sensory neurons in areas 3b and 1 and S2 (Supplementary Fig. 10a–d). Categorical and DPC neurons reach a stationary value at this window-width (Supplementary Fig. 10g, h).

**Periodicity information.** The extended patterns (E) are periodic with a frequency between pulses of 4.34 Hz (pf). Contrary to that, grouped patterns (G) are aperiodic (Fig. 1a, up pattern). Based on the temporal stimulus structure, a phase-locked neuron (sensory) should respond periodically during an extended pattern (E) at 4.34 Hz but not during grouped patterns (G). We aimed to compute the mutual information associated with the pattern identity (G or E) that is conveyed by the periodicity of the neural responses. To accomplish that, we employed Fourier

decomposition of the time signals formed by the evoked trains of spikes during stimulation periods. For each trial, the power spectrum of the spike train evoked during stimulation was computed and normalized. We removed the DC component, so that the total power summed over all positive frequency bins was 100%[14]. Employing this methodological approach, the number of spikes contained in each train had little effect on the resulting Fourier amplitudes, which indicate the proportion of power for each frequency bin. Thus, Fourier amplitudes were mainly determined by the temporal arrangement of the spikes, not by their number. Each trial was first transformed to firing rate employing a quadratic and deterministic kernel of 24 ms and 0.6 ms step. The width of the frequency bins was 0.97 Hz. This value was limited by the duration of the stimulation period, which for the Fourier analysis we took as 1228.8 ms. This means that for each stimulus period we employed 2048 points to compute the Fourier transform, starting 50 ms before and finishing 178.8 ms after the P1 or P2 period.

From each trial, we extracted the two power spectra values associated with the two Fourier frequencies (3.88 Hz and 4.85 Hz) that are nearest to the periodic stimulation frequency (pf = 4.34 Hz). These values should increase for evoked spikes that are more tightly phase-locked to the periodic stimulation pulses. Suppose a neuron is strongly phase-locked to the periodic pattern (E) and fires spikes somewhat like a clock, one or two spikes per stimulus pulse, in an approximately periodic fashion. In its spectra, the maximum power would be at the periodic pattern frequency. Hence, for a sensory neuron, these values should be high during E patterns and small during G patterns.

Similarly, as we explained for categorical information, the mutual information that the periodicity of the response at pf provides about the stimulus pattern (P) is calculated from the probability distributions relating these two variables. The function $P(\text{pf} \mid P)$ represents the conditional probability of observing a spectrum value at pf given that the stimulus pattern had a value of P (G or E). The expression $P(\text{pf})$ describes the probability of observing a spectrum value at pf regardless of the value of the stimulus pattern, and $P(P)$ is the probability that the stimulus takes a value of P (G or E). Then, the information that the spectrum value at pf provides about the pattern identity can be computed as:

$$I_{\text{Per}} = \sum_{\text{pf},P} P(P)P(\text{pf} \mid P)\log_2\left(\frac{P(\text{pf} \mid P)}{P(\text{pf})}\right) \quad (10)$$

For all the mutual information values computed across this work (Eqs. (5)–(10)), a correction for sampling bias was applied[32]. Furthermore, the significance of mutual information values for neurons labeled as sensory and categorical was computed through a permutation test, with the significance criterion set to the $p < 0.01$ level.

**Choice probability**. The choice probability index (CP) was calculated using methods from signal detection theory. In this case, the ROC measures the overlap between hit and error responses for each stimulus pair (P1, P2). A value of 0.5 indicates full overlap, whereas 1 and 0 indicate no overlap between distributions. Thus, the CP quantifies the selectivity for one or the other decision outcome during the discrimination process. To compute the CP as a function of time, we used a window of 200 ms duration moving in steps of 50 ms, beginning at P2 and ending 1500 ms after the animal reported the comparison between P2 and P1. To combine the responses from all neurons at each time bin, the CP values were averaged across all S2 neurons with decision coding (Supplementary Fig. 11a).

**Response latencies**. We calculated two different latencies (Supplementary Fig. 12 and 13): a response latency, which corresponds to the time at which the stimulus-driven neural activity (during P1) becomes significant, and a coding latency, which corresponds to the time at which the encoded signal becomes significant (during P1).

*Response latency*. Firing rate distributions were generated at each time point using a time window of 200 ms sliding steps of 1 ms during P1, and were compared against the rates obtained in a basal period (200 ms before P1 onset) using the ROC method[15]. The first time-bin at which the AUROC was significantly different from 0.5 (permutation test, $p < 0.05$) for five consecutive bins was considered as the response latency to P1.

*Coding latency*. This latency varied depending on the coding profile of the cells. The P1 coding latency was estimated for each neuron by identifying the first of five consecutive bins significantly coding patterns G or E (200 ms windows with 10 ms step).

**Autocorrelation analysis**. The autocorrelation functions of spike counts were computed following the same methodological procedure as in refs. [17,34]. The basal period (−1 to 0 s) was divided into overlapping, successive time bins of 40 ms duration with 20 ms steps. Then, for two time-bins separated by a time lag $t$, we calculated the across-trial correlation between spike counts $N$. Next, we averaged the correlation values computed for each neuron and time lag $t$ across the population. Afterwards, this averaged population autocorrelation function of the time

lag $t$ between bins was fit by an exponential decay with an offset:

$$\text{Aut}(t) = A\left[\exp\left(-\frac{t}{\tau}\right) + B\right] \quad (11)$$

In this equation the autocorrelation tau ($\tau$) measures an intrinsic population timescale. The offset ($B$) represents the contribution of timescales much longer than our observation window. We fit Eq. (11) to the full autocorrelation data from all neurons and trials. Hence, fits were performed at the population level rather than single-neuron level (Supplementary Fig. 14). To be able to fit this equation to the single-neuron level, much more recorded trials per cell are required. To fit Eq. (11) to the population autocorrelation data, a nonlinear least-squares fitting via the Levenberg–Marquardt algorithm was employed.

**Reporting summary**. Further information on research design is available in the Nature Research Reporting Summary linked to this article.

## Data availability
Data files are publicly available at Zenodo (https://doi.org/10.5281/zenodo.4421855); see ref. [48].

## Code availability
The custom MATLAB (R2020b) and C scripts employed in the analysis of this data, as well as the experimental protocols, are available from the corresponding author on reasonable request.

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

## Acknowledgements
We thank H. Diaz and J. Zizumbo for technical assistance. The research of R.R.-P, was partially supported by the Dirección General de Asuntos del Personal Académico de la Universidad Nacional Autónoma de México (PAPIIT-IN210819).

## Author contributions
R.R.-P. and R.R. designed research; A.Z., M.A., and R.R. performed research; R.R.-P., G.D.-d.L. analyzed the data; R.R.-P., G.D.-d.L, and R.R. wrote the paper; and R.R.-P. and R.R. supervised all stages of the study.

## Competing interests
The authors declare no competing interests.
