## [Peer Review File · Nature Communications]

Reviewers' Comments:

Reviewer #1:

Remarks to the Author:

I am satisfied with the revision work.

Reviewer #2:

Remarks to the Author:

I have reviewed a previous version of this manuscript, and it does not seem to have changed dramatically since then so this review is mostly a repetition of the original one.

Animals perform a tactile task consisting of reporting whether two vibrations were the same or different while responses evoked in secondary somatosensory cortex (S2) are recorded. S2 neurons fall into two categories: Neurons that encode the stimulus, much like neurons in S1, and neurons that encode task-relevant variables, for example the trial type (same or different). The sensory responses are the same when the tactile stimuli are no longer behaviorally relevant, but task-related components of the response disappear. The team concludes that S2 plays an active role in transforming sensory inputs into a perceptual decision, and lies between S1 and frontal areas (DPC) that have been previously implicated in sensory decision making.

This is a very nice study with an important result that builds on previous work from this team in a significant way, particularly given the greater complexity of the task.

The study could be improved as follows:

1. It is far too long! The major take-home is actually quite straightforward and is important enough to be reported in a high-impact journal. However, the paper is written for a specialist journal. Results could be presented more parsimoniously by relegating the following components to the supplementary materials for the aficionados:

- a. Some of the rasters in Figures 1 and 2. Pick 2 or 3 that illustrate the main point, relegate the strange cases to the supplement.
- b. The variance analysis in Figures 3C and D. They make only a minor contribution to the general thrust of the paper but contribute to the clutter.
- c. Figures 4 and 5 make a tangential point about the role of spike timing that seems out of place or at least is not integrated well into the general narrative.
- c. In Figure 6, superimpose the three conditions (Hits, errors, LCT, show a subset of conditions (and relegate the others to the supplement). This not only will make this figure more efficient, but it will more effectively get the point across.
- d. In Figure 7, relegate panel A to the Supplement.
- e. All of the latency and autocorrelation could be relegated to the Supplement.

Much of the results obscure the underlying narrative and will only be interesting to folks that are squarely in the field. As such, they could be relegated to the Supplement (including large chunks of the description of the results) and only briefly referenced in the main text.

f. The Discussion is long and rambling, with tangents that do not speak to the data (is the input from S1 or thalamus? e.g. or the bit about consciousness). The Discussion reads more like a Dissertation chapter than a scientific paper. Subheading would be great if they are allowed.

2. No information is provided about the topography of the recorded neurons. As one of the main points of the paper is that there are two distinct groups of neurons (sensory, categorical), one cannot help but wonder whether these two groups are clustered or evenly distributed. Also, there is some suggestion that what is traditionally referred to as S2 is actually two areas S2 and the parietal ventral area (PV). Is it possible that the categorical neurons are in PV?

3. Much of the analysis involves categorizing neurons into one class or the other (sensory or categorical). However, it seems as though neurons really fall into a continuum based on what is shown in Figure 4 (and is briefly mentioned in the Discussion). The classical idea of different categories of neurons that serve different functions seems to have been replaced with an idea that information is multiplexed in the activity of neurons. Perhaps at one extreme there are sensory-like neurons and at

the other extreme there are more cognitive ones, but these are exceptions. Most neurons are a bit of both. While the authors acknowledge this, focusing the analysis on categories of neurons gives the wrong impression. The authors might thus wish to implement population level analyses (dPCA e.g.) to look at multiplexing. Minimally, the authors could de-emphasize the categories, and simply use the categorization as a representation of the extrema of a continuum.

4. Some aspects of the analysis are a bit clunky. For example, the temporal patterns evoke very patterned responses in many of the so-called sensory neurons. However, all of the analyses are rate-based analyses except for the one that places neurons into sensory and non-sensory categories. Given that the stimuli are identical at first, then diverge when the second periodic stimulus occurs, it seems like temporal analyses, where the temporal pattern of activation is analyzed with an expanding window, might be more apt than the firing rate analyses in sliding windows. The conclusions will likely not change, but the analyses will be more elegant. Given this, the coding latency presentation is a bit strange. Indeed, the stimuli are identical at first and the difference only emerges approximately 200 ms in. This is not made explicit in the coding latency section. Given the adopted analysis, the difference can only emerge when the second periodic pulse occurs.

5. The "intrinsic timescale" analysis is not framed well enough. More explanation should be provided about why looking at the autocorrelation is a good measure of this, without the expectation that readers will have read the original Murray et al. paper. It also seems like this point, while potentially interesting, may detract from the main narrative. Furthermore, if the key parameter here is the decay time constant, then S2 and DPC are nearly identical, so it does not drive the point of a hierarchy convincingly.

We have addressed all of the 2nd reviewer's concerns and we appreciate his/her help towards improving the quality of our work. Specific responses to each reviewer's comments are provided below. Based on his/her concerns, we have rewritten the Abstract, Introduction, Results and Discussion sections, with a focus on condensing all these sections and making our narrative clearer. While the main sections (Abstract, Introduction, Results, Discussion and Legends) of the previous version of the manuscript had a total length of 25 pages, the new revised version was reduced to 17 pages, representing an 8-page reduction (without counting References). Particularly, we reduced the introduction from 3.5 pages to 2 pages, the results from 12 to 7.5 pages and the discussion from almost 5 to 3 pages. Moreover, based on his/her recommendations, we moved 3 of the main figures to the supplemental material. Further, we also followed his/her advice and moved several panels of four main figures to the supplemental figures. Additionally, we took out several topics from the discussion, reducing the number of references from 55 to 37. The main sections of the manuscript, including figures and references, are now 24 pages long. Based on all these changes, we believe that our manuscript has been adequately streamlined, and the new manuscript version is a far clearer study. As a result, we believe that the overall message is easier to understand, both for people within and people outside of our particular field. Moreover, we would like to remark that we make emphasis the overwhelming population of the intermediate neurons throughout the revised manuscript. Finally, our revised version includes two novel supplemental figures that were designed to address very particular questions that the reviewer included. In one figure we show that there are not spatial clusters selective for the two distinct type of dynamics (Fig. S6). In another figure (Fig. S9), we show evidence that the optimal integration window-width for decoding pattern identity is ~200ms.

Reviewer #1:

Reviewer: *I am satisfied with the revision work.*

Authors' Closing Response: We are grateful that we were able to answer all his/her concerns. His/her comments were very helpful in improving the quality of our work.

Reviewer #2:

Reviewer: *I have reviewed a previous version of this manuscript, and it does not seem to have changed dramatically since then so this review is mostly a repetition of the original one.*

Authors' response: Although we made our best possible efforts to address his/her concerns in the original revision that was submitted, a great deal of length was necessary to fully elaborate all of the answers that addressed the third reviewer's concerns. We did our best to reduce any section or portion that was not addressed by any of the 3 reviewers, but ultimately these reductions did not overtake the vast additions that we had to include. As we explained above, in the new version of the manuscript, we reduced and condense most of our sections. We hope that this new version satisfy his/her concerns.

Reviewer: *Animals perform a tactile task consisting of reporting whether two vibrations were the same or different while responses evoked in secondary somatosensory cortex (S2) are recorded. S2 neurons fall into two categories: Neurons that encode the stimulus, much like neurons in S1, and neurons that encode task-relevant variables, for example the trial type (same or different). The sensory responses are the same when the tactile stimuli are no longer behaviorally relevant, but task-related components of the response disappear. The team concludes that S2 plays an active role in*

transforming sensory inputs into a perceptual decision, and lies between S1 and frontal areas (DPC) that have been previously implicated in sensory decision making. This is a very nice study with an important result that builds on previous work from this team in a significant way, particularly given the greater complexity of the task.

Authors' response: We are grateful to the reviewer for his/her suggestions to improve our work and for the encouraging comments.

Reviewer: *The study could be improved as follows:*

Authors' response: We hope that the changes we have made based on his/her suggestions appropriately answered his/her concerns.

Reviewer: *It is far too long! The major take-home is actually quite straightforward and is important enough to be reported in a high-impact journal. However, the paper is written for a specialist journal. Results could be presented more parsimoniously by relegating the following components to the supplementary materials for the aficionados.*

Authors' response: We did our best to condense and reduce the manuscript as much as possible. As we explained above, we have rewritten the Abstract, Introduction, Results and Discussion sections, with a focus on condensing all 3. Furthermore, it is our belief that the “straightforward” nature of our narrative is now far more apparent. While the main sections (Abstract, Introduction, Results, Discussion and Legends) of the previous version of the manuscript had a total length of 25 pages, the new revised version was reduced to 17 pages. In particular, we reduced the introduction from 3.5 pages to 2 pages, the results from 12 to 8 pages and the discussion from almost 5 to 3 pages. Then, these 3 sections of the manuscript (Introduction, Results and Discussion) were reduced from 21 to 14 pages. Moreover, based on his/her recommendations, we moved 3 of the main figures to the supplemental material. Further, we also followed his/her advice and moved several panels from four main figures to supplemental figures. Additionally, we took out several topics of the discussion, reducing the number of references from 55 to 37. The main sections of the manuscript, including figures and references, are now 24 pages long.

Reviewer: *a. Some of the rasters in Figures 1 and 2. Pick 2 or 3 that illustrate the main point, relegate the strange cases to the supplement.*

Authors' response: We appreciate his/her suggestion to improve our results. In Fig. 1 we kept only 3 out of 6 neurons in the main figure. In Fig. 2, we reduced the number of neurons from 4 to 2. We relegated all the other neurons to Fig. S1 and Fig. S2.

Reviewer: *b. The variance analysis in Figures 3C and D. They make only a minor contribution to the general thrust of the paper but contribute to the clutter.*

Authors' response: Since we found ourselves in utter agreement with the reviewer, we moved the previous Fig. 3C and D to Fig. S4C and D, respectively. A big portion of the explanation of this figure was moved to the supplemental legend for Fig. S4. Further, the comparison of variance dynamics between S1, S2, and DPC is now far easier to understand. While fast sensory variance dynamics are clear in S2 (Fig. S4C), they vanished, leaving only categorical dynamic in DPC (Fig. S4E) during the

TPDT. Moreover, while variance dynamics vanished in DPC (Fig. S4F), sensory variance remained invariant in S2 (Fig. S4D).

Reviewer: *c. Figures 4 and 5 make a tangential point about the role of spike timing that seems out of place or at least is not integrated well into the general narrative.*

Authors' response: Again, we appreciate his/her recommendation to improve the manuscript. Pursuant to the reviewer's advice, we moved Fig. 5 to the supplemental figures (now Fig. S7). Further, the results' explanations are elaborated in the supplemental legend. With respect to Fig. 4, we demonstrate the differences between sensory and categorical dynamics (Fig. 4C-H). Crucially, it is necessary for the explanation of the selection criteria used to isolated pure categorical and sensory neurons (Fig. 4A-B). However, we reduced this section as much as possible to avoid unnecessary explanations, predominantly to transmit the message more clearly.

Reviewer: *c. In Figure 6, superimpose the three conditions (Hits, errors, LCT, show a subset of conditions (and relegate the others to the supplement). This not only will make this figure more efficient, but it will more effectively get the point across.*

Authors' response: We do really appreciate his/her idea. We superimpose the three conditions in the main figure (Fig. 5). We believe that it is now much simpler to understand. We relegated the complete and old figure to the supplemental material (Fig. S8).

Reviewer: *d. In Figure 6, relegate panel A to the Supplement.*

Authors' response: Again, we appreciate his/her suggestion to improve our results. However, since the majority of the S2 population lies along a continuum of mixing between two pure dynamics, we consider that this figure is relevant to show the change in coding of the entire S2 population depending on the animal's perceptual behavior. For this reason, we have preserved Fig. 6A. Nevertheless, we diminished the length of this section to avoid unnecessary details. We wrote (page 9, 3rd paragraph):

“Even if most neurons with P1 coding (n=1253), are neither pure sensory (n=105) or categorical (n=150), they are highly informative about P1 identity during hit trials (Fig. 6A, blue), including the early part of working memory. However, during error trials these neurons conveyed less information (Fig. 6A, red).”

As an extension, we further clarified (page 9, 3rd paragraph):

“The whole population $I_{P1}(t)$ (Fig. 6A) is a combination between sensory and categorical $I_{P1}(t)$, which likely occurs due to the intermediate neurons. In conclusion, while activity from sensory neurons does not covary with behaviour, categorical responses do; the intermediate responses reflect a balance between dynamics.”

Reviewer: *c. All of the latency and autocorrelation could be relegated to the Supplement. Much of the results obscure the underlying narrative and will only be interesting to folks that are squarely in the field. As such, they could be relegated to the Supplement (including large chunks of the description of the results) and only briefly referenced in the main text.*

Authors' response: Thanks again. We followed his/her recommendation and relegated the latency and autocorrelation figures to be supplemental (Fig. S11 and S13). In the case of the latency results,

while we considered them important, including them in a main figure was ultimately excessive; they can now be found in the supplemental material. Again, most of the latency results' explanations are elaborated in the supplemental legend. In the case of autocorrelation, we also believe that it is an important result, although it is hard to frame its relevance without changing the main focus or our manuscript. Again, most of autocorrelation results were relegated to the legend of the corresponding supplemental figure (Fig. S13).

Reviewer: *The Discussion is long and rambling, with tangents that do not speak to the data (is the input from S1 or thalamus? e.g. or the bit about consciousness). The Discussion reads more like a Dissertation chapter than a scientific paper. Subheading would be great if they are allowed.*

Authors' response: Thanks again! We took out several parts from the discussion. We now focus on our data and results. In particular, we eliminated the paragraph about input from S1 or thalamus and the paragraph about consciousness. We reduced it from 5 pages in the previous version to 3 pages. With respect to the reviewer's proposal, to our knowledge, Nature Communications does not permit subheadings within the discussion section (guide to authors).

Reviewer: *2. No information is provided about the topography of the recorded neurons. As one of the main points of the paper is that there are two distinct groups of neurons (sensory, categorical), one cannot help but wonder whether these two groups are clustered or evenly distributed.*

Authors' response: We appreciate this important concern that allowed us to improve the presentation of our results. To address the reviewer's comment on the spatial clustering or distribution of the pure dynamic subgroups, we have added a supplemental figure that shows the information values for pairs of neurons recorded together (Fig. S6).

Figure S6. Mutual information for pairs of neurons recorded together. (A-C) The information values between neuron 1 and neuron 2 that were recorded simultaneously ($n=3269$ neuron pairs). (A) The $I_{Is,PI}$ values that measure the categorical representation for neuron 1 and neuron 2. The probability of recording a pair of neurons with $I_{Is,PI} > 0.25$ bits is $p=0.013$ ($n=42$). The probability of recording one neuron with $I_{Is,PI} > 0.5$ bits and the other one with $I_{Is,PI} > 0.25$ bits is $p=0.0024$ ($n=8$). No recorded pairs had $I_{Is,PI} > 0.5$ for both neurons. (B) The I_{Per} values used to distinguish the sensory subgroup for neuron 1 and neuron 2. The probability of recording a pair of neighboring neurons with $I_{Per} > 0.25$ bits is $p=0.0067$ ($n=22$). The probability of recording one neuron with $I_{Per} > 0.5$ bits and the other one with $I_{Per} > 0.25$ bits is $p=0.0036$ ($n=12$). No recorded pairs had $I_{Per} > 0.5$ for both neurons. (C) The I_{Per} values of neuron 1 (x axis) plotted against the $I_{Is,PI}$ values of

neuron 2 (y axis). Note that the graph is not the same if the values were to be inverted [$(I_{Per}^1, I_{Is,PI}^2)$ vs. $(I_{Is,PI}^1, I_{Per}^2)$], so we have twice the number of pairs ($n=6538$). The probability of recording a pair of neurons with $I_{Per}^1 > 0.25$ bits and $I_{Is,PI}^2 > 0.25$ bits is $p=0.0078$ ($n=52$). The probability to record one neuron with $I_{Per}^1 > 0.5$ bits (or $I_{Is,PI}^1 > 0.5$ bits) and the other one with $I_{Is,PI}^2 > 0.25$ bits (or $I_{Per}^2 > 0.25$ bits) is $p=0.003$ ($n=19$). No recorded pairs had $I_{Per}^1 > 0.5$ and $I_{Is,PI}^2 > 0.5$ bits. Based on these small probabilities, the different dynamics profiles cannot be spatially segregated into subnetworks. It is highly unlikely to simultaneously record neurons with similar or antagonist dynamics.

Remarkably, paired neurons with high periodicity or categorical coding were unlikely to be recorded simultaneously. Based on this new finding, in the introduction we wrote (page 3, 1st paragraph):

“As an extension, we asked if these distinct coding dynamics depended on two separable subnetworks; however, we found neither spatial segregation based on coding dynamics nor timescales differences across S2. Despite the extreme diversity in S2 coding responses, they appear to develop at the same processing stage.”

In addition, in the results section we wrote (page 7, 2nd paragraph):

“One potential explanation for the two types of dynamics is that they occur in discrete areas within, creating distinct sensory and categorical subnetworks. To address this question, we analyzed the $I_{Is,PI}$ and I_{Per} values conveyed by pairs of neurons recorded together during the TPDT (Fig. S6). We found no clusters in the arrangement of S2 neurons; the probability of recording a pair of nearby neurons with pure dynamic was extremely low.”

Our findings show that the pure dynamics, as well as intermediate dynamics, must be found distributed with relative equality across the entire S2 network, meaning that there is no actual subnetwork formed within S2 that could be attributed with the isolation of certain dynamics. Further, in Fig. S13 we found that there is no segregation across their timescales. In the discussion (page 13, 3rd paragraph) we wrote:

“Motivated by the clear differences between sensory and categorical responses, we wondered whether these contrasting dynamics arise within differentiable subnetworks. It is unlikely to record pairs of pure sensory or categorical neurons in nearby electrodes, indicating that neurons with similar responses do not appear in spatial clusters. Further, the intrinsic timescale of neural fluctuations, estimated with the autocorrelation decay constant, increases from sensory to frontal lobe cortices in monkeys¹⁶ and mice³². We found the same timescale for all S2 responses, regardless of dynamic profile. Across the somatosensory hierarchy, S1 displays fast timescales and phase-locked responses, while S2 and DPC exhibit much longer time constants that are appropriate for temporal integration. Further, the optimal integration window to decode pattern identity from 3b and S2 sensory responses was ~200ms, which concurs with the S2 timescale found here. Moreover, categorical and DPC neurons saturate their coding capacity at around the same windows wide. Moreover, categorical and DPC neurons saturate their coding capacity at around the same window width. All this evidence taken together does not support the notion of subnetworks or structural differences.”

Reviewer: Also, there is some suggestion that what is traditionally referred to as S2 is actually two areas S2 and the parietal ventral area (PV). Is it possible that the categorical neurons are in PV?

Authors' response: We appreciate his/her concern. We returned to the protocols from our recording sessions to assure ourselves that all recordings occurred squarely within the S2 region, such that the

neurons could not possibly be found in PV instead of S2. In the methods section we wrote (page 3, 2nd paragraph, Supplemental Information and Methods):

“We were careful to record just above the primary auditory cortex (A1), and we tested this using auditory stimuli to ensure that the neurons were only responding to vibrotactile stimuli. The receptive fields of the recorded neurons were all very large and some were bimanual, and since the monkey’s hand was carefully fixed in the same manner during each recording session, we do not believe it is possible for these neurons to be responding to motor data in a categorical manner, as would be seen in the parietal ventral area (PV). Concurrently, categorical decision responses during P2 or after pb disappeared during the LCT (Fig. 4).”

For further clarification, we added (page 3, 3rd paragraph, Supplemental Information and Method):

“Recording sites changed from session to session; the locations of the penetrations were used to construct surface maps in S1, S2 and DPC by marking the edges of the small chamber (7 mm in diameter) placed above each area. It is important to emphasize that the sensory and categorical neuron subgroups were both recorded across the entire S2 region. The probability of recording two pure subgroup responses together is extremely low (Fig. S6), so we did not record sufficient pure pairs for further analyses.”

Reviewer: 3. *Much of the analysis involves categorizing neurons into one class or the other (sensory or categorical). However, it seems as though neurons really fall into a continuum based on what is shown in Figure 4 (and is briefly mentioned in the Discussion). The classical idea of different categories of neurons that serve different functions seems to have been replaced with an idea that information is multiplexed in the activity of neurons.*

Authors’ response: As we explained before, we have rewritten our manuscript, condensing and emphasizing the critical role that the predominant intermediate neurons play within the processing network. The reviewer’s interpretation is both astute and correct; as opposed to significant subgroups of neurons serving distinct functions, the majority of the network is multiplexing either the sensory or categorical information across a wide breadth of intermediate neurons, such that the information is still parallel in its dynamics to the corresponding pure dynamics subgroup. We employed the extreme neurons to analyze and characterize the purest responses, knowing that most of the population mixes these dynamics. Based on that, we consider that Fig. 4 is extremely important to exhibiting that sensory and categorical neurons are the extreme responses among the population. Further in the abstract we wrote:

“Importantly, the vast majority of the neurons fall along a continuum of combined sensory and categorical dynamics.”

To clarify, we included this question at the beginning of the introduction (page 2, 1st paragraph):

“Do these responses appear as a continuum between distinct neural codes? Would these distinct neural codes be associated with separable subnetworks?”

Additionally, towards the end of the introduction, we added (page 3, 1st paragraph):

“Further, the S2 population reflected a range of intermediate dynamics that varied between pure sensory and pure categorical; the vast majority of the S2 network falls along this continuum of

combinations. Moreover, across the S2 responses, categorical information increased during the TPDT with respect to the LCT, and sensory information diminished. Consequently, which information is predominant in the whole S2 population strongly depends on task context.”

Further, in the section “**A continuum from phase-lock to categorical neurons**”, we wrote (page 7, 2nd paragraph):

“Notice that a comparable number of sensory (n=105) and categorical (n=150) neurons were identified with our criteria (I_{Per} or $I_{Is,PI}>0.25$ Bits, see Fig. S5A). Remarkably, no neurons were found along the diagonal that satisfied both criteria (Fig. 4C). Neurons not classified as sensory or categorical represented the brunt of the population, exhibiting low or intermediate values for both types of information (green points). Notably, when using the same metrics for the neurons recorded during the LCT (n=313, Figs. 4D), the plots exhibited drastic changes. In the LCT, almost all neurons with high information were sensory neurons (Fig. 4D); categorical neurons were reduced drastically.”

Reviewer: *Perhaps at one extreme there are sensory-like neurons and at the other extreme there are more cognitive ones, but these are exceptions. Most neurons are a bit of both. While the authors acknowledge this, focusing the analysis on categories of neurons gives the wrong impression.*

Authors’ response: As this relates to the previous concern, we agree with the reviewer, and in the revised version we emphasized the intermediate results. We rewrote the section where we analyze 313 neurons that were recorded in both conditions (TPDT and LCT); since most of these neurons are intermediate, we center our discussion on the implications of this population. However, there is a shift in the information conveyed by these neurons (page 11):

“We inquired whether mutual information values depended on the cognitive context (TPDT or LCT), so we compared the same metrics in a subgroup of neurons recorded during both tasks (n=313, Fig. S7). Specifically, we wondered whether single neurons changed the type of information conveyed depending on the task condition. Each neuron represents a point in Fig. S7A, defined by the TPDT I_{Per} (x axis) and the LCT I_{Per} (y axis). The angle distribution between the two axes was biased to higher values ($\langle\theta\rangle=57.49^\circ$), meaning that neurons have a higher degree of phase-locking responses during LCT than TPDT. In contrast, neurons displayed larger values of categorical information ($I_{Is,PI}$) during the TPDT than the LCT ($\langle\theta\rangle=31.45^\circ$, Fig. S7B). Summarily, periodicity information tends to grow during LCT and categorical information increases during the TPDT. In agreement, several exemplary neurons with intermediate responses increase their sensory response by decreasing their categorical coding during LCT (Fig. S2B-E).”

Reviewer: *The authors might thus wish to implement population level analyses (dPCA e.g.) to look at multiplexing.*

Authors’ response: Due to the previous studies we have published in DPC, we knew that the population-level results that could be extracted using dPCA could be extremely useful to understand the role of intermediate neurons^{1,2}. Moreover, analyzing the intermediate dynamics in detail would require the number of necessary figures to increase appreciably. However, the current version of the manuscript has 7 main figures and 13 supplemental figures. As the reviewer’s initial comment regarded length, we considered that this addition would undermine our efforts to condense our manuscript. That being said, we added a paragraph to the Discussion about population analyses within S2 or across simultaneously recorded areas (page 20, 2nd paragraph):

“Recently, we have analyzed the heterogeneous responses observed in frontal lobe neurons with dimensionality reduction techniques¹⁻³, which allow us to condense the network’s signals, preserving the significant population dynamics. Notably, for each component, the weights given to each neuron occurred with a Gaussian like distribution^{1,2}. These continua of responses are parallel to the gradient of intermediate dynamics we have shown in S2, so a promising line of inquiry would be the application of such techniques to interpret mixed responses at a population level. In another recent study, analogous methods were applied to investigate the simultaneous recordings of two visual sensory areas, V1 and V2⁴. They revealed a population level mechanism in V1 that influences a small part of the total activity fluctuations observed in V2. The somatosensory network could be implementing a similar mechanism for the routing of sensory information from S1 to S2 and categorical information to DPC. Future experiments and analyses are required to answer this hypothesis.”

Reviewer: *Minimally, the authors could de-emphasize the categories, and simply use the categorization as a representation of the extrema of a continuum.*

Authors’ response: We have changed our manuscript to frame the role of these extreme responses appropriately. We hope that these changes address this particular concern.

Reviewer: *4. Some aspects of the analysis are a bit clunky. For example, the temporal patterns evoke very patterned responses in many of the so-called sensory neurons. However, all of the analyses are rate-based analyses except for the one that places neurons into sensory and non-sensory categories. Given that the stimuli are identical at first, then diverge when the second periodic stimulus occurs, it seems like temporal analyses, where the temporal pattern of activation is analyzed with an expanding window, might be more apt than the firing rate analyses in sliding windows. The conclusions will likely not change, but the analyses will be more elegant.*

Authors’ response: We appreciate his/her concern a lot, and we think that the reviewer has noticed a very important point. Following his/her recommendation, in the revised version of the manuscript, we employed an expanding window to show that sensory neurons exhibit a clear optimal integration window at around 200ms width. On the other hand, categorical neurons display a stationary response that saturate at around the same window-width. The whole S2 neurons depict an intermediate behavior. While sensory S2 neurons exhibit a behavior similar to area 3b, categorical neurons behave analogously to DPC neurons. These results are shown in a new supplemental figure (Fig. S9):

Figure S9. Optimal integration window across cortices and subgroups of neurons. (A- H) We compute the mean pattern information carried by different groups of neurons, measured in bits (Methods) as a function of window size (10-1000ms in steps of 10ms) during P1 [light traces] or P2 [dark traces]. We calculated the mutual information ($I(P1,r)$ and $I(P2,r)$) for each neuron using time windows of different lengths. Then, we averaged the information across all neurons to obtain the mean population information for each window. (A-D) Subgroups of neurons or areas with phase locking responses exhibit an information maximum when the windows was ~ 200 ms wide. Notably, the peak decrease significantly from area 3b (A) in comparison with the whole S1 population (B). Sensory S2 neurons (C-D) depict a comparable behaviour than area 3 neurons (A) and a much higher maximum than S1 population (B). Further, sensory S2 neurons exhibit higher $I(P,r)$ during LCT (D) than TPDT (C). (E-H) In the other populations of neurons (E, G, H), information reached a stationary value at around 200ms. In agreement with Fig. 3, in panel F, information exhibited a noteworthy decrease from the whole S2 population (E). (G) Subpopulation of categorical neurons computed during P1 ($n=150$). Analogous results was found with P2 categorical neurons. (H) DPC neurons depict much smaller P2 information. In summary, while sensory neurons exhibit a clear optimal integration window, categorical neurons reach a stationary value at around the same value, ~ 200 ms.

Given the actual length of the manuscript, we do not have enough space to discuss this result properly. However, on page 9, 2nd paragraph we wrote:

“First, we normalized the activity (z-score) at each time bin from the 1253 neurons with significant P1 coding (200ms window, 50ms step, Methods). A 200ms window was optimal for decoding P1 identity from areas 3b, 1 and S2 sensory neurons (Fig. S9). In particular, the information carried by categorical and DPC neurons saturates at this windows-width.”

Furthermore, in the Discussion section, we mention the relation between the optimal integration window and the autocorrelation timescale found in the S2 network (page 13, 3rd paragraph):

“Further, the optimal integration window to decode pattern identity from 3b and S2 sensory responses was ~ 200 ms, which concords with the S2 timescale found here. Moreover, categorical and DPC neurons, saturate their coding capacity at around the same windows wide.”

Reviewer: *Given this, the coding latency presentation is a bit strange. Indeed, the stimuli are identical at first and the difference only emerges approximately 200 ms in. This is not made explicit in the coding latency section. Given the adopted analysis, the difference can only emerge when the second periodic pulse occurs.*

Authors' response: Again, we appreciate the reviewer's concern for clarity and improvement of the presentation of our results. As we explained above, we moved the latency figure and most of its results to the supplementary material. However, to clarify, we included an explanatory sentence on the legend of the corresponding supplemental figure (Fig. S11):

“Expectedly, the average coding latency across each population was longer than their corresponding response latencies. Based on experimental design, since the patterns became distinct with the arrival of the 2nd E-pattern pulse (~ 220 ms), the coding latency must be longer than this value; the fastest coding latency across all S2 neurons was never less than 230ms.”

Reviewer: *5. The “intrinsic timescale” analysis is not framed well enough. More explanation should be provided about why looking at the autocorrelation is a good measure of this, without the*

expectation that readers will have read the original Murray et al. paper. It also seems like this point, while potentially interesting, may detract from the main narrative.

Authors' response: We appreciate his/her recommendation greatly. As we explained above, the autocorrelation figure is now supplementary, with the results being primarily found in the corresponding supplemental legend. Further, we removed any mention of timescale from the abstract. In the introduction, we also rewrote the paragraph related with this findings, remarking that it was employed to study if the sensory and categorical neurons exhibited a segregation in timescales (page 3, 1st paragraph):

“As an extension, we asked if these distinct coding dynamics depended on two separable subnetworks; however, we found neither spatial segregation based on coding dynamics nor timescales differences across S2. Despite the extreme diversity in S2 coding responses, they appear to develop at the same processing stage.”

Further, the autocorrelation results were condensed and integrated with the latency results, resulting in a single section that refers to both findings (page 10):

“S2 in the somatosensory hierarchy: latencies and inherent timescales”

“Moreover, in recent works, timescales of intrinsic fluctuations across cortices were presented within a hierarchical framework¹⁶, using the autocorrelation function. We applied this metric to each subgroup and to the entire network of S2 (Fig. S13A). Surprisingly, we observed analogous autocorrelation decay rates for the whole S2 population ($\tau=178\text{ms}$), as well as for sensory ($\tau=182\text{ms}$) and categorical neurons ($\tau=187\text{ms}$). Even if sensory and categorical neurons exhibit completely different latencies, their autocorrelations are similar. These results support the idea that, although different in function, S2 subpopulations are embedded within the same processing stage (see Fig. S6).”

In relation, we changed the focus of the discussion paragraph (page 13, 3rd paragraph):

“Motivated by the clear differences between sensory and categorical responses, we wondered whether these contrasting dynamics arise within differentiable subnetworks. It is unlikely to record pairs of pure sensory or categorical neurons in nearby electrodes, indicating that neurons with similar responses do not appear in spatial clusters. Further, the intrinsic timescale of neural fluctuations, estimated with the autocorrelation decay constant, increases from sensory to frontal lobe cortices in monkeys¹⁶ and mice³². We found the same timescale for all S2 responses, regardless of dynamic profile. Across the somatosensory hierarchy, S1 displays fast timescales and phase-locked responses, while S2 and DPC exhibit much longer time constants that are appropriate for temporal integration. Further, the optimal integration window to decode pattern identity from 3b and S2 sensory responses was $\sim 200\text{ms}$, which concords with the S2 timescale found here. Moreover, categorical and DPC neurons, saturate their coding capacity at around the same windows wide. Moreover, categorical and DPC neurons saturate their coding capacity at around the same window width. All this evidence taken together does not support the notion of subnetworks or structural differences.”

Reviewer: *Furthermore, if the key parameter here is the decay time constant, then S2 and DPC are nearly identical, so it does not drive the point of a hierarchy convincingly.*

Authors' response: The key parameter, the tau decay constant, is parallel between S2 and DPC, but the key distinction we have highlighted in the text is that the autocorrelation values are far higher for

DPC, throughout the autocorrelation time lag possibilities. This means that even though the activity may be reverberating within the networks for a similar amount of time, the strength of the reverberation is very different. Notably, both areas exhibit a timescale of around 200ms, which we showed (Fig. S9) to be the optimal integration window. In the supplemental legend (Fig. S13), we wrote:

“Despite comparable τ values between the S2 and DPC populations, the autocorrelation results for DPC start at and maintain greater values, meaning fluctuations do not reverberate equally within each network, despite approximately equivalent timescales.”

Authors’ Closing Response: We acknowledge sincerely the reviewer’s enthusiasm for our results, and we are thankful for the chance to improve the quality of our manuscript immensely, especially in terms of condensing and streamlining our narrative. Summarily, we really appreciate your support.

References:

1. Rossi-Pool, R. *et al.* Decoding a Decision Process in the Neuronal Population of Dorsal Premotor Cortex. *Neuron* **96**, 1432-1446.e7 (2017).
2. Rossi-Pool, R. *et al.* Temporal signals underlying a cognitive process in the dorsal premotor cortex. *Proc. Natl. Acad. Sci.* **116**, 7523–7532 (2019).
3. Romo, R. & Rossi-Pool, R. Turning Touch into Perception. *Neuron* **105**, 16–33 (2020).
4. Semedo, J. D., Zandvakili, A., Machens, C. K., Yu, B. M. & Kohn, A. Cortical Areas Interact through a Communication Subspace. *Neuron* **102**, 249-259.e4 (2019).
5. Rossi-Pool, R. *et al.* Emergence of an abstract categorical code enabling the discrimination of temporally structured tactile stimuli. *Proc. Natl. Acad. Sci.* **113**, E7966–E7975 (2016).
6. Murray, J. D. *et al.* A hierarchy of intrinsic timescales across primate cortex. *Nat. Neurosci.* **17**, 1661–1663 (2014).
7. Wang, X.-J. Macroscopic gradients of synaptic excitation and inhibition in the neocortex. *Nat. Rev. Neurosci.* **21**, 169–178 (2020).
8. Siegle, J. H. *et al.* A survey of spiking activity reveals a functional hierarchy of mouse corticothalamic visual areas. *bioRxiv* 805010 (2019) doi:10.1101/805010.

Reviewers' Comments:

Reviewer #2:

Remarks to the Author:

Authors did a great job addressing my comments!

One significant omission, however, is that they fail to report how the neuronal sample was distributed over the two monkeys: How many of the 1646 neurons were tested in each monkey with TPDT task and how many of the 313 neurons in each monkey were tested with the LCT and TPDT tasks? Were results from the two monkeys identical? Given the relative complexity of the task, this is a relevant point.

We have addressed the 2nd reviewer's concerns and we appreciate his/her help towards improving the quality of our work. Based on his/her concerns, we have incorporated a new supplemental figure where we show the performance and the neural coding dynamics for both monkeys (Fig. S5). Furthermore, throughout the manuscript we have specified the number of neurons, sessions, performance, sensory and categorical neurons from each monkey. The instances where details were added throughout the manuscript are all highlighted in yellow. Importantly, all results are highly consistent between monkeys. Although neural coding dynamics exhibit some differences, the main results endure. Based on that observation, in the previous versions of the manuscript we had decided not to include this figure. However, as the reviewer pointed out, it is a relevant information given the complexity of our task. Specific responses to the reviewer's comments are provided below.

Reviewer #2:

Reviewer: *Authors did a great job addressing my comments!*

Authors' response: We are extremely grateful to the reviewer for his/her suggestions to improve our work and for the encouraging comments. In our revised version of the manuscript, we have incorporated a new supplemental figure where we showed performance and neural coding dynamics for both monkeys.

Reviewer: *One significant omission, however, is that they fail to report how the neuronal sample was distributed over the two monkeys: How many of the 1646 neurons were tested in each monkey with TPDT task and how many of the 313 neurons in each monkey were tested with the LCT and TPDT tasks?*

Authors' response: We appreciated this reviewer's concern, since it helped us to comprehend the relevance of this piece of information. We have incorporated the details about the number of neurons, sessions, sensory and categorical neurons recorded in each monkey. On page 4 we wrote:

“We recorded extracellular activity from 1646 neurons in S2 (Fig. 1C, Methods) during the monkeys' performance of the TPDT (Monkey RR17, n=1035; Monkey RR20, n=611).”

“Several of the S2 neurons recorded during the TPDT were also recorded during the LCT (n=313; Monkey RR17, n=189; Monkey RR20, n=124),....”

On page 7 we wrote:

“Notice that a comparable number of sensory (n=105; Monkey RR17, n=71; Monkey RR20, n=34) and categorical (n=150; Monkey RR17, n=91; Monkey RR20, n=59) neurons were identified with our criteria (I_{per} or $I_{Is,PI} > 0.25$ Bits, see Fig. S6A).”

Further, in the legend of the new Fig. S5 we wrote:

“Figure S5. Performance and population coding dynamics during TPDT and LCT across Monkeys. (A) Monkey RR17's performance for the whole TPDT (84.5%, grey, n_{SES}=281 sessions), for each class [85.6% G-G (red), 83.1% G-E (orange), 83.5% E-G (green), 85.9% E-E (blue)] and for the whole light control task (LCT) (100%, yellow, n_{SES}=49 sessions). In the LCT, the same stimuli were delivered as in the TPDT, but the rewarding push button press was visually guided. (B) Monkey

RR20's performance for the whole TPDT (83.1%, $n_{SES}=142$ sessions), for each class: G-G (81.5%), G-E (82.6%), E-G (83.7%), E-E (84.6%) and for LCT (100%, $n_{SES}=27$ sessions). (C) To facilitate comparison, we included performance from both monkeys' sessions (same as Fig. 1A) during TPDT (84.0%, $n_{SES}=423$ sessions), G-G (84.2%), G-E (82.9%), E-G (83.6%), E-E (85.4%) and LCT (100%, $n_{SES}=76$ sessions). (A-B) Note the consistency across monkeys' performance. (D-G) Percentage of neurons with significant coding (see Fig. S3) as a function of time during TPDT (Monkey RR17, $n=1035$; Monkey RR20, $n=611$) and LCT (Monkey RR17, $n=189$; Monkey RR20, $n=124$). Traces refer to P1 (cyan), P2 (green), all class coding (pink), and decision coding (black). Despite analogous results between both monkeys during the TPDT, ranging through the comparison period, P2 coding is higher in RR17 and class coding is more prominent in RR20. In both monkeys, P1 working memory, decision, and class coding, essentially vanished during the LCT: instead, the coding was restricted to stimulus periods. Similar to DPC, all categorical and perceptual codes are abolished during LCT, but akin to area 3b, S2 sensory responses always persist (see Fig. 7). Further, in both monkeys we identified sensory neurons ($I_{Per}>0.25$ bits, Monkey RR17, $n=71$; Monkey RR20 $n=34$) and categorical neurons computed during the first stimulus ($I_{Is,P1}>0.25$ bits, Monkey RR17, $n=91$; Monkey RR20, $n=59$) or second stimulus ($I_{Is,P2}>0.25$ bits, Monkey RR17, $n=116$; Monkey RR20, $n=68$) periods."

In the Methods section (page 17), we wrote:

"We computed the average performance across S2 recording sessions ($p=84.0\%$; Monkey RR17, $p=84.5\%$ and Monkey RR20, $p=83.1\%$). 1B and Fig. S5A-C."

"After training in the TPDT, the monkeys saturated their average performance around 84% (Fig. 1B and Fig. S5A-C, $n_{SES}=423$ recording sessions; Monkey RR17, $n_{SES}=281$; Monkey RR20, $n_{SES}=142$)."

"However, the performance for the LCT was consistently 100% (Fig. 1B and Fig. S5A-C, $n_{SES}=76$ recording sessions; Monkey RR17, $n_{SES}=49$; Monkey RR20, $n_{SES}=27$);"

Finally, in the Methods section (page 18), we also wrote:

"We recorded 1646 S2 neurons using the TPDT stimulus set with 5 Hz mean frequency (Monkey RR17, $n=1035$; Monkey RR20, $n=611$). Additionally, we have a dataset of $n=313$ neurons (Monkey RR17, $n=189$; Monkey RR20, $n=124$) that were tested in both the LCT and in TPDT using the 5Hz mean frequency set."

Reviewer: *Were results from the two monkeys identical? Given the relative complexity of the task, this is a relevant point.*

Authors' response: Again, we appreciate this recommendation greatly since it will help to clarify the consistency of our results across monkeys.

As we state above, the performance was consistent across monkeys. Even if the task is extremely complex, we trained both animals until they reached approximately the same, stable performance in the whole task and across classes (Fig. S5A-C).

Further, we have analyzed the neural coding dynamics for each monkey during TPDT and LCT in detail. We have included these results in the new Fig. S5D-G. The main difference across monkeys is that P2 coding is more prominent in Monkey RR17 and class coding is more abundant in Monkey RR20. The decision coding after pu is more abundant in Monkey RR17. P1 coding is a little higher in Monkey RR20. The number of sensory neurons and coding during LCT are analogous:

Figure S5. Performance and population coding dynamics during TPDT and LCT across Monkeys.

Reviewers' Comments:

Reviewer #2:

Remarks to the Author:

I am satisfied. Good work!

We have responded to the 2nd reviewer's commentary.

Reviewer #2:

Reviewer: *I am satisfied. Good work!*

Authors' Closing Response: We are pleased at having been able to address all of the reviewer's concern, as well as appreciate of his encouraging response. His/her comments were very essential to improving our manuscript throughout this review process, and we would like to sincerely thank all of their constructive commentary.